# Revisiting the Variational Information Bottleneck

## Abstract

The Information Bottleneck (IB) framework offers a theoretically optimal approach to data modeling, though it is often intractable. Recent efforts have optimized supervised deep neural networks (DNNs) using a variational upper bound on the IB objective, leading to enhanced robustness to adversarial attacks. In these studies, supervision assumes a dual role: sometimes as a presumably constant and observed random variable, and at other times as its variational approximation. This work proposes an extension to the IB framework, and consequently to the derivation of its variational bound, that resolves this duality. Applying the resulting bound as an objective for supervised DNNs induces significant empirical improvements, and provides an information theoretic motivation for decoder regularization.

## 1 Introduction

The Variational Information Bottleneck, VIB, (Alemi et al. 2017) adapts the theoretically optimal[1], yet mostly intractable, Information Bottleneck, IB, (Tishby et al. 1999) to supervised DNNs. However, the IB is a method for unsupervised learning, and requires knowledge of the underlying joint distribution $p(x, y)$ (Slonim 2002). This requirement is relaxed in the original VIB derivation, resulting in a duality in the usage of the downstream RV $Y$, which is treated both as an observed RV when sampled from the training data, and as a variational approximation when optimized over. This work proposes a new adaptation of the IB and VIB frameworks for supervised tasks, and consequently an information-theoretic motivation for decoder regularization.

We begin by laying down what IB is, and how it can be adapted to DNNs. Classic information theory provides rate-distortion (Shannon 1959) for optimal compression of data. However, rate-distortion regards all information as equal, not taking into account which information is more relevant to a specified downstream task, without constructing tailored distortion functions. The Information Bottleneck (IB) (Tishby et al. 1999) resolves this limitation by defining mutual information (MI) between the learned representation and a designated downstream random variable (RV) as a universal distortion function. Yet, learning representations using the IB method is possible given discrete distributions, and some continuous ones, but not in the general case (Chechik et al. 2003). Moreover, MI is either difficult or impossible to optimize over when considering deterministic models, such as

---

[1]Optimal data modeling with the IB method is established under the assumption that optimizing a precision-complexity trade-off will yield a model that is closer in nature to the real underlying process, and that mutual information is a sufficient metric for this purpose (Slonim 2002).

DNNs (Saxe et al. 2018; Amjad & Geiger 2020). Nonetheless, the promise of the IB remains alluring, and recent efforts utilized VAE (Kingma & Welling 2014) inspired variational methods to approximate upper bounds on the IB objective, allowing its utilization as a loss function for DNNs, where the underlying distributions are both continuous and unknown (Alemi et al. 2017; Fischer 2020; Cheng et al. 2020). These approaches learn representations in supervised settings, without knowledge of the underlying distribution $p(x, y)$, utilizing the learned variational conditional $p(y|x)$ to approximate MI. In contrast, non variational IB methods learn representations in unsupervised settings, where the stochastic process underlying the observed data is known (Tishby et al. 1999; Chechik et al. 2003; Painsky & Tishby 2017). Nonetheless, when deriving the variational IB objectives, previous research considered the learned representation as the only optimized RV, when in practice a variational classifier is also optimized. This work proposes a modification of the IB and variational IB objectives, by setting the downstream RV as a parameterized model in the problem definition. We believe our modification is a better adaptation of the IB for supervised tasks, and show empirical evidence of improved performance across several challenging tasks over different modalities. Finally, we use our findings to propose a novel information theoretic interpretation of overfitting in supervised DNNs.

The reader is encouraged to refer to the preliminaries provided in Appendix A before proceeding.

## 2 RELATED WORK

### 2.1 DETERMINISTIC INFORMATION BOTTLENECK

Classic information theory offers rate-distortion (Shannon 1959) to mitigate signal loss during compression: A source $X$ is compressed to an encoding $Z$, such that maximal compression is achieved while keeping the encoding quality above a certain threshold. Encoding quality is measured by a task specific distortion function: $d : X \times Z \mapsto \mathbb{R}^+$. Rate-distortion suggests a mapping that minimizes the rate of bits to source sample, measured by $I(X; Z)$, that adheres to a chosen allowed expected distortion $D \geq 0$. The Information Bottleneck (IB) (Tishby et al. 1999) extends rate-distortion by replacing the tailored distortion functions with MI over a target distribution: Let $Y$ be the target signal for some specific downstream task, such that the joint distribution $p(x, y)$ is known, and define the distortion function as MI between $Z$ and $Y$. The IB is the solution to the optimization problem $Z : \min_{p(z|x)} I(X; Z)$ subject to $I(Z; Y) \geq D$, that can be optimized by minimizing the IB objective $\mathcal{L}_{IB} = \beta I(X; Z) - I(Z; Y)$ over $p(z|x)$. The solution to this objective is a function of the Lagrange multiplier $\beta$, and is a theoretical limit for representation quality, given mutual information as an accepted metric, as elaborated in more detail in Appendix B. The IB is in fact an unsupervised soft clustering problem, where each data point $x$ is assigned a probability $z$ to belong to different clusters, given the joint distribution of the input and target tasks $p(x, y)$ (Slonim 2002). Chechik et al. (2003) showed that computing the IB for continuous distributions is hard in the general case, and provided a method to optimize the IB objective in the case where $X, Y$ are jointly Gaussian and known. Painsky & Tishby (2017) offered a limited linear approximation of the IB for any distribution by extracting the jointly Gaussian element of given distributions. Saxe et al. (2018) considered the application of the IB objective as a loss function for DNNs, and concluded that computing mutual information in deterministic DNNs is problematic as the entropy term $H(Z|X)$ for a continuous $Z$ is infinite. Amjad &

Geiger (2020) extended this observation and pointed out that for a discrete $Z$ MI becomes a piecewise constant function of its parameters, making gradient descent limited and difficult.

Considering the supervised problem, Geiger & Fischer (2020) suggested to consider the classification output as an additional random variable, leading to an extended Markov chain underlying the problem: $Y \leftrightarrow X \leftrightarrow Z \leftrightarrow \tilde{Y}$. A similar approach has also been suggested by Piran et al. (2020) where a dual IB formulation was suggested that although still considers the minimization of $I(X; Z)$ replaces the constraints to one that takes into account $\tilde{Y}$. The approach suggested here follows these ideas, but adds the additional objective of reducing overfitting during the classification step.

## 2.2 Variational Information Bottleneck

Alemi et al. (2017) introduced the Variational Information Bottleneck (VIB) - a variational approximation for an upper bound to the IB objective for DNN optimization. Bounds for $I(X, Z)$ and $I(Z, Y)$ are derived from the non negativity of KL divergence, and are used to form an upper bound for the IB objective. Variational approximations are then used to replace intractable distributions in the upper bound. Using the *reparameterization trick* (Kingma & Welling 2014), a discrete empirical estimation of the variational upper bound is used as a loss function for classifier DNN optimization, resulting in a loss function that is equivalent to the $\beta$-autoencoder loss (Higgins et al. 2017). VIB was evaluated over image classification tasks, and displayed substantial improvements in robustness to adversarial attacks, while inflicting a slight reduction in test set accuracy, when compared to equivalent deterministic models. The improved robustness is attributed to an improvement in representation quality, and subsequently better generalization. Achille & Soatto (2018) extended VIB with a total correlation term, designed to increase latent disentanglement. Fischer (2020) proposed an IB based loss function named Conditional Entropy Bottleneck (CEB), in which the conditional mutual information of $X$ and $Z$ given $Y$ is minimized, instead of the unconditional mutual information. The CEB loss, $L_{CEB} = \min_{Z} I(X; Z|Y) - \gamma I(Y; Z)$, is designed to minimize all information in $Z$ that is not relevant to the downstream task $Y$, by conditioning over $Y$. CEB is equivalent to IB for $\gamma = \beta - 1$ following the chain rule of mutual information (Cover 1999) and the IB Markov chain, as established in Appendix B. However, its variational approximation, VCEB, differs from VIB in the way the marginal is approximated. Geiger & Fischer (2020) showed that VCEB is a tighter variational approximation for IB under certain conditions, but not in the general case. Later work (Fischer & Alemi 2020) evaluated VCEB on the ImageNet-A and ImageNet-C datasets, two flavors of ImageNet (Deng et al. 2009) that assess model performance on challenging edge cases and robustness to common corruptions, respectively. Results showed improved generalization, calibration, and robustness to the targeted PGD (Madry et al. 2018) attack, in particular when model size was increased.

## 2.3 Information theoretic regularization

Label smoothing (Szegedy et al. 2016) and entropy regularization (Pereyra et al. 2017) regularize classifier DNNs by increasing classifier entropy, either by inserting a scaled conditional entropy term to the objective, or by smoothing the training labels. Applying either methods improved test accuracy and model calibration on various challenging tasks. Alemi et al. (2018) extended the information plane (Tishby et al. 1999) to VAE (Kingma & Welling 2014) settings, measuring distortion as MI between input and reconstructed images,

and rate as KL divergence between variational representation and marginal. The limits of representation quality in VAEs are looser than the theoretical IB limits, and heavily depend on the chosen variational families of the marginal and decoder distributions. The closer the families are to the true distributions, the tighter the gap to the theoretical IB limit for representation quality. Alemi et al. (2018) also showed that the ELBO loss is prone to produce low quality representations: provided a strong enough decoder, the ELBO KL regularization term might induce completely uninformative representations, that are then overfitted by the powerful decoder, as elaborated in detail in Appendix B. In the current study, a conditional entropy term (Pereyra et al. 2017) emerges during the derivation of our proposed adaptation of the IB objective, providing a possible remedy to the discrepancies in the ELBO loss, and subsequently VIB and VCEB loss, described in (Alemi et al. 2018).

## 3 FROM VIB TO SVIB

### 3.1 PROBLEM DEFINITION

As elaborated in Section 2.1, the IB objective, $\mathcal{L}_{IB} = I(X;Z) - \beta I(Z;Y)$, is computed over the joint distribution $p(x,y,z)$. When $p(x,y)$ is given, this expression is optimized over the distribution $p(z|x,y)$, as proposed by Tishby et al. (1999):

$$\min_{p(z|x,y)} I(Z;X)$$
$$\text{s.t.} \quad I(Z;Y) \geq D_1 \tag{1}$$

However, as mentioned, adapting IB to supervised tasks admits the learned classifier as a new RV to the optimization problem (Geiger & Fischer 2020; Piran et al. 2020). Thus, we consider the extended Markov chain $Y \leftrightarrow X \leftrightarrow Z \leftrightarrow \tilde{Y}$ for supervised IB, distinguishing between the true unknown RV $Y$, and the learned classifier $\tilde{Y}$. We follow this approach, and also assume that $\tilde{Y}$ and $Y$ share the same support. The IB framework connects the underlying joint distribution of the input and objective data, $p(x,y)$, with a learned representation $Z$. We claim that when applying IB to supervised tasks, one must also consider the connection to the classifier defined by the output RV $\tilde{Y}$. Thus, we also want to consider the joint distribution over the pair $Z, \tilde{Y}$ during optimization. Following the IB method logic, we seek a $\tilde{Y}$ that will minimize mutual information with $Z$, whilst keeping below a defined distortion metric with the true $Y$. That is, we seek a second bottleneck that minimizes passage of information between $Z$ and $\tilde{Y}$, so as to limit it to the minimum required to ensure that $\tilde{Y}$ is similar enough to $Y$, given the transition through both $X$ and $Z$. Since in this case we are optimizing over the joint conditional distribution $p(z,\tilde{y}|x,y)$, instead of the conditional $p(\tilde{y}|z,x,y)$, this problem is not simply an IB problem over the Markov chain $Y \leftrightarrow Z \leftrightarrow \tilde{Y}$. Moreover, contrary to the standard IB, $X$ plays a significant role, controlling the distribution of $Z$, and the entire chain of four random variables must be taken into consideration. We thus define a second bottleneck for the true distribution $p(x,y)$ and modeled distribution $c(\tilde{y}|z)p(z|x,y)$. We choose KL divergence as a distortion metric, as we assume $Y$ and $\tilde{Y}$ share the same support. For some positive scalar $D_2$ we have:

$$\min_{c(\tilde{y}|z)p(z|x,y)} I(Z;\tilde{Y})$$
$$\text{s.t.} \quad D_{KL}\left(p(y=\tilde{y}|z,x) \middle\| c(\tilde{y}|z)p(z|x)\right) \leq D_2 \tag{2}$$

Combining the two bottlenecks results in a new optimization problem, which we denote Supervised Information Bottleneck (SIB), which minimizes the following objective:

$$\mathcal{L}_{SIB} \equiv \beta I(X;Z) - I(Z;Y) + \lambda I(Z;\tilde{Y}) + D_{KL}\left(p(y = \tilde{y}|z, x)\Big\|c(\tilde{y}|z)p(z|x)\right) \qquad (3)$$

## 3.2 Optimization Objective

We proceed to derive a tractable variational upper bound for $\mathcal{L}_{SIB}$, which we can use as an objective function for classifier DNNs. We begin by deriving the first bottleneck (1) as done in VIB (Alemi et al. 2017), and proceed to derive the second (2).

Consider $I(Z;X)$:

$$I(Z;X) = \int\int p(x,z)\log\left(p(z|x)\right)\mathrm{d}x\,\mathrm{d}z - \int p(z)\log\left(p(z)\right)\mathrm{d}z \qquad (4)$$

For any probability distribution $r$ we have that $D_{KL}\left(p(z)\big\|r(z)\right) \geq 0$, it follows that:

$$\int p(z)\log\left(p(z)\right)\mathrm{d}z \geq \int p(z)\log\left(r(z)\right)\mathrm{d}z \qquad (5)$$

And so, by Equation 5:

$$I(Z;X) \leq \int\int p(x)p(z|x)\log\left(\frac{p(z|x)}{r(z)}\right)\mathrm{d}x\,\mathrm{d}z \qquad (6)$$

Consider $I(Z;Y)$:

From the Barber-Agakov inequality (Barber & Agakov 2003), we have that for any probability distribution $c$:

$$I(Z;Y) \geq \int\int p(y,z)\log\left(c(y|z)\right)\mathrm{d}y\,\mathrm{d}z - \int p(y)\log\left(p(y)\right)\mathrm{d}y \qquad (7)$$

Note that Equations 6 and 7 hold for any distribution $r$ over the support of $Z$, and for any conditional distribution $c(\cdot|z)$ whose support equals the support of $Y$ for every given value $z$ in the support of $Z$. We link the two bottlenecks by choosing $c$ to be $\tilde{Y}|Z = z \sim c(\cdot|z)$, meaning the variational classifier distribution. This connection is implicit in (Alemi et al. 2017), where $\tilde{Y}$ is not formally defined. We now move on to the second bottleneck.

Consider $I(Z;\tilde{Y})$:

$$I(Z;\tilde{Y}) = H(\tilde{Y}) - H(\tilde{Y}|Z) \qquad (8)$$

Choosing a discrete random variable for $\tilde{Y}$, as in labeled classification, we have $H(\tilde{Y}) \leq \log\|\tilde{\mathcal{Y}}\|$. Otherwise, choosing a continuous RV with finite support $[a, b]$, we have that $H(\tilde{Y}) \leq \log(b - a)$. In both cases, $I(Z;\tilde{Y})$ is bounded from above by some constant

$J = \log(b - a)$, or $J = \log \| \tilde{Y} \|$, and the negative conditional entropy term $-H(\tilde{Y}|Z)$:

$$I(Z; \tilde{Y}) \leq J - H(\tilde{Y}|Z) = J + \int \int p(\tilde{y}, z) \log\left(c(\tilde{y}|z)\right) d\tilde{y} \, dz \tag{9}$$

Consider $\underline{D_{KL}\left(p(y = \tilde{y}|z, x) \middle\| c(\tilde{y}|z)p(z|x)\right)}$:

$$D_{KL}\left(p(y = \tilde{y}|z, x) \middle\| c(\tilde{y}|z)p(z|x)\right) = \tag{10}$$

$$\int \int \int p(y, z, x) \log\left(p(y|z, x)\right) dy \, dx \, dz - \int \int \int p(y, z, x) \log\left(c(y|z, x)\right) dy \, dx \, dz \tag{11}$$

Applying the Markov chain $Y \leftrightarrow X \leftrightarrow Z \leftrightarrow \tilde{Y}$, and total probability, we get:

$$D_{KL}\left(p(y = \tilde{y}|z, x) \middle\| c(\tilde{y}|z)p(z|x)\right) =$$
$$\int \int p(y, x) \log\left(p(y|x)\right) dy \, dx - \int \int p(y, z) \log\left(c(y|z)\right) dy \, dz \tag{12}$$

Finally, we attain an upper bound for $\mathcal{L}_{SIB}$ by combining Equations (6,7,9,12):

$$\mathcal{L}_{SIB} \leq \beta \int \int p(x)p(z|x) \log\left(\frac{p(z|x)}{r(z)}\right) dx \, dz - 2 \int \int p(y, z) \log\left(c(y|z)\right) dy \, dz$$
$$+ \lambda \int \int c(y|z)p(z) \log\left(c(y|z)\right) dy \, dz + \int \int p(y, x) \log\left(p(y|x)\right) dy \, dx$$
$$+ \int p(y) \log\left(p(y)\right) dy + \lambda J \tag{13}$$

Note that $p(x, y)$ and $J$ are constants, and so the last three terms in Equation (13) can be ignored in the course of optimization.

### 3.3 Variational approximation and empirical estimation

We further develop the upper bound in Equation (13) using the IB Markov chain $Y \leftrightarrow X \leftrightarrow Z \leftrightarrow \tilde{Y}$ and total probability, and define tractable variational distributions to replace intractable ones. Let $e(z|x)$ a variational encoder approximating the conditional $p(z|x)$, let $r(z)$ be a variational approximation for the marginal, and let let $c(y|z)$ a variational classifier approximating $p(y|z)$. We define the variational approximation $L_{SVIB}$:

$$\mathcal{L}_{SVIB} \equiv \beta \int \int p(x)e(z|x) \log\left(\frac{e(z|x)}{r(z)}\right) dx \, dz$$
$$- 2 \int \int \int p(x)p(y|x)e(z|x) \log\left(c(y|z)\right) dx \, dy \, dz$$
$$+ \lambda \int \int \int p(x)e(z|x)c(y|z) \log\left(c(y|z)\right) dx \, dy \, dz \tag{14}$$

As is common in VIB and VAE literature, we chose a standard Gaussian for the variational marginal $r(z)$, a spherical Gaussian for the variational encoder $e(z|x)$, and a categorical distribution for the variational classifier $c(y|z)$. We use DNNs to model these distributions as follows: Let $e_\phi(z|x) \sim N(\mu, \Sigma)$ be a stochastic DNN encoder with parameters $\phi$, and a final layer of dimension $2K$, such that for each forward pass, the first $K$ entries are used to encode $\mu$, and the last $K$ entries to encode a diagonal $\Sigma$, after a soft-plus transformation. Let $C_\gamma$ be a discrete classifier neural net parameterized by $\gamma$, such that $C_\gamma(y|z) \sim Categorical$. $r(z)$ is constant and unparameterized. We use Monte Carlo sampling over some discrete dataset $\mathcal{S}$ to empirically estimate $\mathcal{L}_{SVIB}$. The true and possibly continuous distribution $p(x, y) = p(y|x)p(x)$ can be sampled from $\mathcal{S}$. Distributions featuring $Z$ are samples from the stochastic encoder using the *reparameterization trick* (Kingma & Welling 2014), such that for each $x_n \in \mathcal{S}$ we generate a sample $\hat{z}_n$. Finally, we use the variational classifier to attain instances $\tilde{y}_n$, given an instance $\hat{z}_n$.

$$\widehat{\mathcal{L}}_{SVIB} \equiv \frac{1}{N} \sum_{n=1}^{N} \left[ \beta D_{KL}\left( e_\phi(z|x_n) \middle\| r(z) \right) - \log\left( C_\gamma\left(y_n|\hat{z}_n\right)\right) + \lambda \log\left(C_\gamma(\tilde{y}_n|\hat{z}_n)\right) \right] \quad (15)$$

### 3.4 Motivation

Tishby et al. (1999) proposed that representations are optimal if they contain just enough information for a required downstream task, and proposed the information bottleneck as a method to obtain such representations. However, in the supervised case an additional information processing stage is added, where representations are decoded by a learned decoder[2], in a joint training process. As mentioned in Section 2.3, Alemi et al. (2018) observed that the ELBO loss function (Kingma & Welling 2014) may learn uninformative representations even when strong KL regularization is imposed, since an overpowerful decoder can overfit the learned embeddings. This observation holds for all VIB loss functions (Alemi et al. 2017; Fischer 2020; Cheng et al. 2020), as VIB is equivalent to the ELBO loss, as shown in (Alemi et al. 2017). Our proposed extension to the IB and VIB frameworks asks to resolve this conflict. By appending an additional bottleneck between the representation $Z$, and learned classifier $\tilde{Y}$, we learn a classifier that holds the minimal information about the representation that is required to meet a designated distortion target over the true downstream RV. Extending the work in (Alemi et al. 2018). We propose to define a decoder $\tilde{Y}$ as overfitting, if a substantial amount of its information about $Z$ lacks relevance about $Y$. The conditional MI $I(Z; \tilde{Y}|Y)$ measures the amount of information $Z$ and $\tilde{Y}$ share, that is uninformative about about $Y$. Hence, we have that $\tilde{Y}$ overfits $Z$ if:

$$I(Z; \tilde{Y}) \gg I(Z; \tilde{Y}) - I(Z; \tilde{Y}|Y)$$
$$H(\tilde{Y}|Y) \gg H(\tilde{Y}|Z) \quad (16)$$

Where the last line follows from the SIB Markov chain.

By deriving the second bottleneck, $\mathcal{L}_{SVIB}$ introduces a modulated conditional entropy term to the loss function: $-\lambda H(\tilde{Y}|Z)$, inducing an increase in the right hand side of Equation 16. At the same time, we expect that the left hand side conditional entropy will be reduced, by

---

[2]Here decoder in the general sense, including classifiers and other decoders

the power of the cross entropy term. Applying these two forces together prevents decoders from overfitting embeddings, as is illustrated in Figure 1.

Figure 1: Venn diagrams illustrating decoder overfitting. The left diagram depicts an overfitted decoder where $\tilde{Y}$ holds no information about $Y$, and $H(\tilde{Y}|Y) \gg H(\tilde{Y}|Z)$. The right diagram depicts a regularized decoder where $H(\tilde{Y}|Y)$ is not much greater than $H(\tilde{Y}|Z)$.

## 4 EXPERIMENTS

We follow the experimental setup proposed by Alemi et al. (2017), extending it to NLP tasks as well. We trained image classification models on the ImageNet 2012 dataset (Deng et al. 2009), and text classification models on the IMDB sentiment analysis dataset (Maas et al. 2011). For each dataset, we compared a competitive Vanilla model with VIB models trained over 8 different $\beta$ values ranging from $10^{-4}$ to 0.5, a VCEB model trained with $\rho$ values ranging from 1 to 7, and an SVIB model trained with different combinations of $\beta$ and $\lambda$ values. All models were trained over a frozen encoder of the vanilla model, to allow faster experimentation. Each model was trained and evaluated 5 times per setting. Models were evaluated over test set accuracy and robustness to various adversarial attacks, showing consistent performance. For image classification, we employed the untargeted Fast Gradient Sign (FGS) attack (Goodfellow et al. 2015), as well as the targeted CW $L_2$ attack (Carlini & Wagner 2017), (Kaiwen 2018). For text classification, we used the untargeted Deep Word Bug attack (Gao et al. 2018), (Morris et al. 2020) as well as the untargeted PWWS attack (Ren et al. 2019). The empirical results presented in Figure 2 confirms that while VIB, VCEB and SVIB models mostly decrease test set accuracy compared to the vanilla model, they significantly improve robustness to the applied adversarial attacks. SVIB attains significantly higher test set accuracy over VIB and VCEB, notably outperforming the vanilla model for IMDB, while scoring the highest robustness in all attacks, apart from the CW attack. A comparison of the best VIB, VCEB and SVIB models further substantiates these findings, with statistical significance confirmed by a p-value of less than 0.05 on a Wilcoxon rank sum test. We note that our experiments compare identical models, varying only in objective functions and scaling parameters. This design highlights performance differences solely due to these factors. Methods like training from scratch to boost overall performance were omitted to ensure a robust comparison. Elaboration on the experimental setup, detailed results, and further insights from the experiments are available in Appendix C. Code to reconstruct the experiments is provided in the supplementary materials of this submission.

## 4.1 IMAGE CLASSIFICATION

A pre-trained inceptionV3 (Szegedy et al. 2016) base model was used and achieved a 77.21% accuracy on the ImageNet 2012 validation set (Test set for ImageNet is unavailable). Image classification evaluation results are shown in Figure 2, examples of successful attacks are shown in Figures 6, 7 in Appendix C.

## 4.2 TEXT CLASSIFICATION

A fine tuned BERT uncased (Devlin et al. 2019) base model was used, and achieved a 93.0% accuracy on the IMDB sentiment analysis test set. Text classification evaluation results are shown in Figure 2, examples of successful attacks are shown in Figures 1,2 in Appendix C.

**Comparison of IMDB and IMAGENET Results Across Models and Metrics**

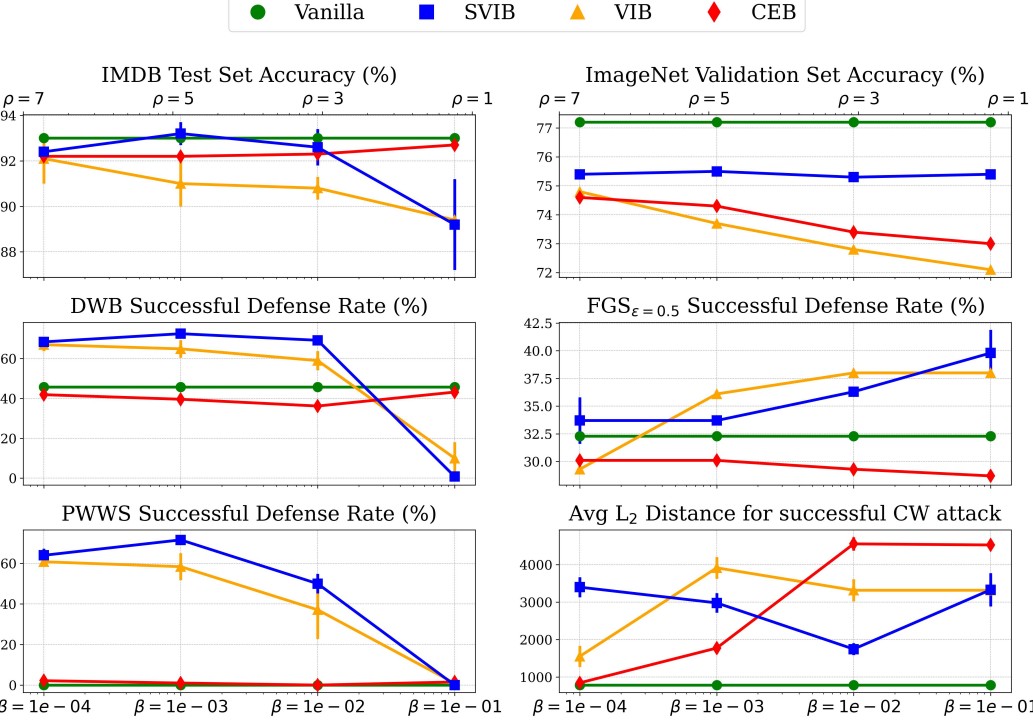

Figure 2: Performance comparison across models and metrics for IMDB and ImageNet. **Higher is better ↑ in all plots**. Analyzing accuracy and robustness against adversarial attacks for vanilla, SVIB, VIB, and VCEB models under Varying $\beta$ and $\rho$ values, average over 5 runs with standard deviation. Left column features IMDB tasks, right column features ImageNet tasks. Upper row shows accuracy over test set, and bottom rows depict robustness under various adversarial attacks, presented as the rate of deflected attacks, or as the average $L_2$ distance required for a successful CW attack. Results show that SVIB attains significantly higher test set accuracy, outperforming the vanilla model for IMDB, while attaining better robustness in all attacks apart from the CW attack. $\rho$ values apply to CEB models, while $\beta$ values apply for SVIB and VIB models. SVIB results are presented for $\lambda = 1$ in IMDB and $\lambda = 2$ in ImageNet. For all experimental results please see the results Section in Appendix C.

## 5 Discussion

The IB is a special case of rate-distortion, and was initially designed to optimize compressed representations. Applying the IB objective for supervised tasks results in optimization of a classifier distribution as well, and requires a reformulation of the initial problem to include both representation and classification. We propose Supervised IB (SIB), an extension to the original IB that considers the classifier distribution as well, and adds an additional bottleneck to mitigate information flow between representations and classifier. We derive a tractable variational approximation for SIB, SVIB, and show that it outperforms VIB and VCEB in terms of classification accuracy and robustness to adversarial attacks, over high dimensional tasks of different modalities, with high statistical significance. We use previous information theoretic frameworks for deep learning (Alemi et al. 2018; Pereyra et al. 2017; Szegedy et al. 2016) to interpret our findings, and propose a definition for decoder overfitting, and a new motivation for conditional entropy regularization. While other advancements have been achieved in recent years, (Fischer 2020; Cheng et al. 2020; Achille & Soatto 2018), none propose a reformulation for IB, as is required in our opinion.

This study opens many opportunities for further research: Applying SVIB in self-supervised learning, and in particular measuring whether representations learned with SVIB capture better semantics than representations learned with non IB inspired loss functions, empirical studies with a full covariance matrix SVIB, a GMM model SVIB, adding $\beta$ and $\lambda$ annealing to SVIB, and combining SVIB with CEB are left for future work.

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

## APPENDIX A - PRELIMINARIES

### NOTATION

We denote random variables (RVs) with upper cased letters $X, Y$, and their realizations in lower case $x, y$. Denote discrete Probability Mass Functions (PMFs) with an upper case $P(x)$ and continuous Probability Density Functions (PDFs) with a lower case $p(x)$. Subscripts are written where the RVs identities are not clear from the context, and hat notation denotes empirical measurements.

Let $X, Y$ be two observed random variables with a true and unknown joint distribution $p(x, y)$, and true marginals $p(x)$, $p(y)$. We can attempt to approximate these distributions using a model $p_\theta$ with parameters $\theta$, such that for generative tasks $p_\theta(x) \approx p(x)$, and for discriminative tasks $p_\theta(y|x) \approx p(y|x)$, using a dataset of $N$ i.i.d observation pairs $\mathcal{S} = \{(x_1, y_1), ..., (x_N, y_N)\}$ to fit our model. One can also assume the existence of an additional unobserved RV $Z \sim p(z)$ that influences or generates the observed RVs $X, Y$. Since $Z$ is unobserved, it is absent from the dataset $\mathcal{S}$, and so cannot be modeled directly. Denote $p_\theta(x) = \int p_\theta(x|z) p_\theta(z) \, \mathrm{d}z = \int p_\theta(x, z) \, \mathrm{d}z$ the marginal, $p_\theta(z)$ the prior as it is not conditioned over any other RV, and $p_\theta(z|x)$ the posterior following Bayes' rule.

### VARIATIONAL APPROXIMATIONS

When modeling an unobserved variable of an unknown distribution, we encounter a problem as the marginal $p_\theta(x) = \int p_\theta(x, z) \, \mathrm{d}z$ doesn't have an analytic solution. This intractability can be overcome by choosing some tractable parametric variational distribution $q_\phi(z|x)$ to approximate the posterior $p_\theta(z|x)$, such that $q_\phi(z|x) \approx p_\theta(z|x)$, and estimate $p_\theta(x, z)$ or $p_\theta(x, z|y)$ by fitting the dataset $\mathcal{S}$ (Kingma & Welling 2019).

### LEARNING TASKS

Vapnik (1995) defines *supervised* learning as follows:

- A generator of random vectors $x \in \mathbb{R}^d$, drawn independently from an unknown probability distribution $p(x)$.

- A supervisor who returns a scalar output value $y \in \mathbb{R}$, according to an unknown conditional probability distribution $p(y|x)$. We note that these probabilities can indeed be soft labels, where $y$ is a continuous probability vector, rather the more commonly used hard labels.

- A learning machine capable of implementing a predefined set of functions, $f(x, \theta) : \mathbb{R}^d \times \Theta \mapsto \mathbb{R}$, where $\Theta$ is a set of parameters.

The problem of supervised learning is that of choosing from the given set of functions, the one that best approximates the supervisor's response, based on observation pairs from the training set $\mathcal{S}$, drawn according to $p(x, y) = p(x)p(y|x)$.

Slonim (2002) defines *unsupervised* learning as the task of constructing a compact representation of a set of unlabeled data points $\{x_1, ..., x_N\}, x_i \in \mathbb{R}^d$, which in some sense reveals their hidden structure. This representation can be used further to achieve a variety of goals, including reasoning, prediction, communication etc. In particular, *unsupervised clustering* partitions the data points into exhaustive and mutually exclusive clusters, where each cluster

can be represented by a centroid, typically a weighted average of the cluster's members. *Soft clustering* assigns cluster probabilities for each data point, and fits an assignment by minimizing the expected loss for these probabilities, usually a distance metric such as MSE.

INFORMATION THEORETIC FUNCTIONS

In this work, information theoretic functions share the same notation for discrete and continuous settings, and are denoted as follows:

| | Notation | Differential | Discrete |
|---|---|---|---|
| **Entropy** | $H_p(X)$ | $-\int p(x) \log (p(x)) \, \mathrm{d}x$ | $-\sum_{x \in X} P(x) \log (P(x))$ |
| **Conditional entropy** | $H_p(X\|Y)$ | $-\int \int p(x,y) \log (p(x\|y)) \, \mathrm{d}x \, \mathrm{d}y$ | $-\sum_{x \in X} \sum_{y \in Y} P(x,y) \log (P(x\|y))$ |
| **Cross entropy** | $CE(p,q)$ | $-\int p(x) \log (q(x)) \, \mathrm{d}x$ | $-\sum_{x \in X} P(x) \log (Q(x))$ |
| **Joint entropy** | $H_p(X,Y)$ | $-\int \int p(x,y) \log (p(x,y)) \, \mathrm{d}x \, \mathrm{d}y$ | $-\sum_{x \in X} \sum_{y \in Y} P(x,y) \log (P(x,y))$ |
| **KL divergence** | $D_{KL}\left(p\|\|q\right)$ | $\int p(x) \log \left(\frac{p(x)}{q(x)}\right) \, \mathrm{d}x$ | $\sum_{x \in X} P(x) \log \left(\frac{P(x)}{Q(x)}\right)$ |
| **Mutual information (MI)** | $I(X;Y)$ | $\int \int p(x,y) \log \left(\frac{p(x,y)}{p(x)p(y)}\right) \, \mathrm{d}x \, \mathrm{d}y$ | $\sum_{x \in X} \sum_{y \in Y} P(x,y) \log \left(\frac{P(x,y)}{P(x)P(y)}\right)$ |

## Appendix B - Related work elaboration

This appendix supplements the related work presented in Section 2, by providing a deeper review of the IB, the IB theory of deep learning, and variational approximations for the IB.

### The information plane

As mentioned in Section 2.1, the solution to the IB objective, $\mathcal{L}_{IB} = I(X; Z) - \beta I(Z; Y)$, depends on the Lagrange multiplier $\beta$. Hence, the IB objective has no one unique solution, and can thus be plotted as a function of $\beta$ and of $Z$'s cardinality, over a Cartesian system composed of the axes $I(X; Z)$ (rate) and $I(Z; Y)$ (distortion). We denote the resulting curve the *information curve*, and its Cartesian system the *information plane* (Tishby et al. 1999), as illustrated in Figure 3. When $\beta$ approaches 0 the distortion term is nullified and we learn a representation that has maximal compression but no information over the down stream task (such a representation may be a null vector), and when $\beta$ approaches $\infty$ we learn a representation that has the maximal possible information over the downstream task, but minimal compression. The region above the information curve is unreachable by any possible representation. The different bifurcation of the information curve, illustrated in Figure 3, correspond to the different possible cardinalities of the compressed representation.

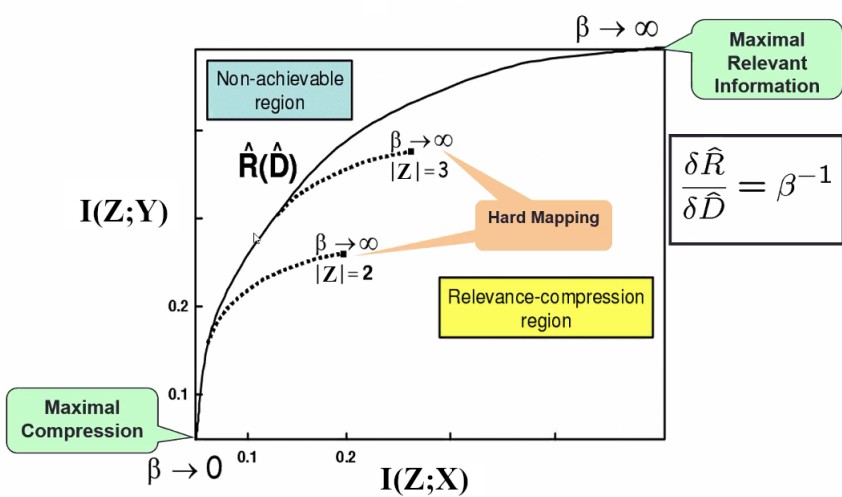

Figure 3: The information plane and curve: rate-distortion ratio over $\beta$. At $\beta = 0$ the representation is compressed but uninformative (maximal compression), at $\beta \to \infty$ the representation is informative but potentially overfitted (maximal information). Taken from (Slonim 2002).

### Fixing a Broken ELBO

Kingma & Welling (2014) introduced variational auto encoders (VAEs) as a latent model based generative DNN architecture. In VAEs, an unobserved RV $Z$ is assumed to generate evidence $X$, a variational DNN encoder $e(z|x)$ is used to approximate the intractable posterior $p(z|x)$, and a variational DNN decoder $d(\hat{x}|z)$ is used to reconstruct $X$. The log probability $log\,(p(x))$ is developed in to the tractable Evidence Lower Bound (ELBO) loss: $log\,(p(x)) \leq \mathcal{L}_{\text{ELBO}}(x) \equiv -\mathbb{E}_{e(z|x)}\left[log\,(d(x|z))\right] + D_{KL}\left(e(z|x)\big|\big|m(z)\right)$, consisting of a

reconstruction error term (cross entropy), and a KL regularization term between encoder and variational marginal $m(z)$.

Alemi et al. (2018) adapt the information plane (Tishby et al. 1999) to VAEs by defining an additional theoretical bound for the ratio between rate and distortion, imposed by the limits of finite parametric families of variational approximations. Instead of true rate and distortion, the proposed information plane features variational rate as $R \equiv D_{KL}\left(e(z|x)\|m(z)\right)$, and variational distortion as $D \equiv -\int\int p(x)e(z|x)log\left(d(x|z)\right)dxdz$. Figure 4 illustrates the suggested information plane, which is divided into three sub planes: (1) Infeasible: This is the IB theoretical limit (As per Figure 3); (2) Feasible: Attainable given an infinite model family, and complete variety of $e(z|x), d(x|z)$ and $m(z)$; (3) Realizable: Attainable given a finite parametric and tractable variational family. The black diagonal line at the lower left satisfies $H_p(X) - D = R$, resulting in tight variational bounds on the mutual information.

Alemi et al. (2018) observe that the variational rate $R$ does not depend on the variational decoder distribution $d(x|z)$. As $R$ is used as the ELBO KL regularizer, high variational compression rates can be attained regardless of MI between decoder and learned representation. Equivalently, good reconstruction does not directly depend on good representation. Empirical evidence suggest that VAEs are prone to learn uninformative representations while still achieving low ELBO loss, a degeneration made possible by overpowerful decoders that are able to overfit the little information captured by the encoder. $D_{KL}\left(e(z|x)\|m(z)\right)$ approaches 0 iff $e(z|x) \rightarrow m(z)$, making $e(z|x)$ close to independence from $x$, resulting in a latent representation that fails to encode information about the input. However, a suitably powerful decoder could possibly learn to overfit encoded traces of the training examples, and reach a low distortion score during optimization.

In the current study, we extend this theoretical framework to explain the advancements of our proposed loss function.

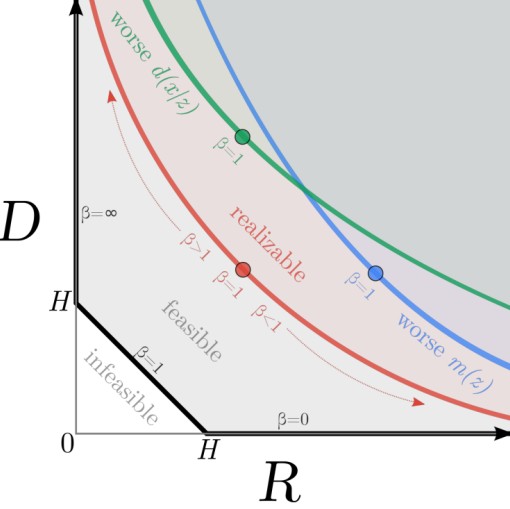

Figure 4: Phase diagram, a proposed information plane interpretation of VAEs. Axes are variational rate and distortion. The IB theoretical limit is extended by an additional limit induced by the constraint of a finite parametric variational family. Once a family is chosen, we seek to learn an optimal marginal $m(z)$ and decoder $d(x|z)$ in order to approach the new limit. Taken from (Alemi et al. 2018).

IB THEORY OF DEEP LEARNING

The following is a summary of work leveraging the IB framework for deterministic DNN optimization and interpretation. For a more comprehensive review of this opinion-splitting topic, the reader is advised to consult the work of Goldfeld & Polyanskiy (2020).

Tishby & Zaslavsky (2015) proposed a representation-learning interpretation of DNNs using the IB framework, regarding DNNs as Markov cascades of intermediate representations between hidden layers. Under this notion, comparing the optimal and the achieved rate-distortion ratios between DNN layers will indicate if a model is too complex or too simple for a given task and training set. Shwartz-Ziv & Tishby (2017) visualized and analyzed the information plane behavior of DNNs over a toy problem with a known joint distribution. Mutual information of the different layers was estimated and used to analyze the training process. The learning process over Stochastic Gradient Descent (SGD) exhibited two separate and sequential behaviors: A short Empirical Error Minimization phase (ERM) characterized by a rapid decrease in distortion, followed by a long compression phase with an increase in rate until convergence to an optimal IB limit, as demonstrated in Figure 5. Similar, yet repetitive behavior was observed in the current study, as elaborated in Section 5.

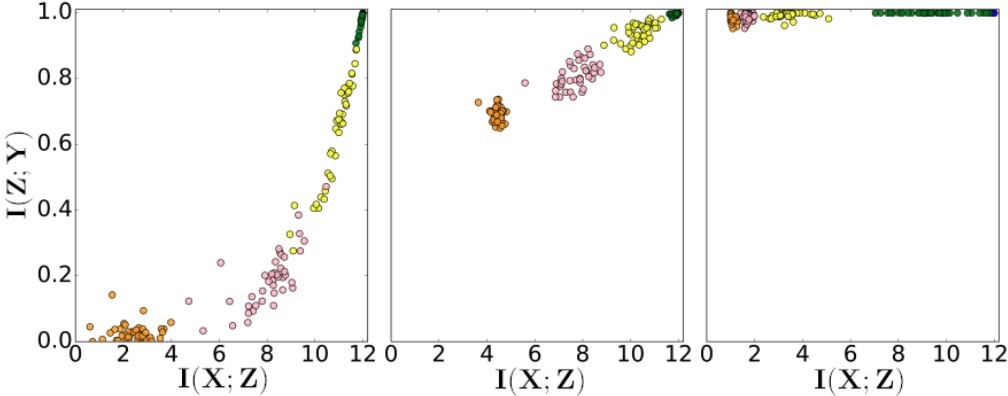

Figure 5: Information plane scatters of different DNN layers (colors) in 50 randomized networks. Left are initial weights, center are at 400 epochs, and right at 9000 epochs. Taken from Shwartz-Ziv & Tishby (2017).

Saxe et al. (2018) reproduced the experiments described in (Shwartz-Ziv & Tishby 2017), expanding them to different activation functions, different datasets and different methods to estimate mutual information. It was found that double-sided saturated nonlinear activations, such as the tanh, produced a distinct compressions stage when mutual information was measured by binning, as performed in (Shwartz-Ziv & Tishby 2017), while other activations did not. It was also shown that DNN generalization did not depend on a distinct compression stage, and that DNNs do forget task irrelevant information, but this happens concurrently to the learning of task relevant information, and not necessarily separately. Amjad & Geiger (2020) argued against the use of the IB as an objective for deterministic DNNs, as mutual information in deterministic DNNs is either infinite or step like, because of mutual information's invariance to invertible transformations, and because of the absence of a decision function in the objective. Using IB as an objective in stochastic DNNs, such as of the variational IB family, is suggested as a possible solution. When examining the

information plane behavior in the current study, we notice recurring patterns of distortion reduction followed by rate increase, resembling the ERM and representation compression stages described by Shwartz-Ziv & Tishby (2017), as elaborated in Appendix 5.

CONDITIONAL ENTROPY BOTTLENECK

As mentioned in Section 2.2, Fischer (2020) showed that the conditional entropy bottleneck is equivalent to IB for $\gamma = \beta - 1$ following the chain rule of mutual information (Cover 1999), and the IB Markov chain. We develop this equivalence in detail:

$$
\begin{aligned}
CEB =& I(X;Z|Y) - \gamma I(Z;Y) \\
\overset{\text{MI chain rule}}{=}& H(Z|Y) - H(Z|X,Y) - \gamma I(Z;Y) \\
\overset{Z \leftarrow X \leftrightarrow Y}{=}& H(Z|Y) - H(Z|X) - \gamma I(Z;Y) \\
\overset{\gamma := \beta - 1}{\Longrightarrow}& H(Z|Y) - H(Z|X) - (\beta - 1)I(Z;Y) \\
=& H(Z|Y) - H(Z|X) - \beta I(Z;Y) + I(Z;Y) \\
=& H(Z|Y) - H(Z|X) - \beta I(Z;Y) + H(Z) - H(Z|Y) \\
=& H(Z) - H(Z|X) + H(Z|Y) - H(Z|Y) - \beta I(Z;Y) \\
=& I(X;Z) - \beta I(Z;Y)
\end{aligned}
$$

## APPENDIX C - EXPERIMENTS ELABORATION

Image classification models were trained on the first 500,000 samples of the ImageNet 2012 dataset (Deng et al. 2009), and text classification over the entire IMDB sentiment analysis dataset (Maas et al. 2011). For each dataset, a competitive pre-trained model (Vanilla model) was evaluated and then used to encode embeddings. These embeddings were then used as a dataset for a new stochastic classifier net with either a VIB or a SVIB loss function. Stochastic classifiers consisted of two ReLU activated linear layers of the same dimensions as the pre-trained model's logits (2048 for image and 768 for text classification), followed by reparameterization and a final softmax activated FC layer. Learning rate was $10^{-4}$ and decaying exponentially with a factor of 0.97 every two epochs. Batch sizes were 32 for ImageNet and 16 for IMDB. All models were trained using an Nvidia RTX3080 GPU with approximately 1-2 days per a single experiment run. Beta values of $\beta = 10^{-i}$ for $i \in \{1, 2, 3\}$ were tested, and we used a single forward pass per sample for inference, since previous studies indicated that these are the best range and sample rate for VIB (Alemi et al. 2017; 2018). Each model was trained and evaluated 5 times per $\beta$ value, with consistent performance. Statistical significance was demonstrated in all comparisons using the Wilcoxon rank sum test with all metrics compared attaining a p-value of less than 0.05. Rank sum was computed as follows: A sorted vector of results was prepared for each compared metric, where each entry featured the attained result in each of the 5 i.i.d. experiments per algorithm, and a boolean indicator value for the algorithm type. For example, let $r :=$ $((0.94, 1) \, (0.935, 1) \, (0.93, 1) \, (0.93, 1) \, (0.925, 1) \, (0.92, 0) \, (0.915, 0) \, (0.915, 0) \, (0.91, 0) \, (0.89, 0))$ be a sorted vector of (test accuracy, algorithm) tuples, 1 being SVIB, 0 VIB. We compute the rank-sum as follows:

$$\mu_T = \frac{5 \cdot 11}{2} = 27.5, \;\; \sigma_T = \sqrt{\frac{5 \cdot 5 \cdot 11}{12}} \approx 4.78, \;\; Z(T) = \frac{15 - 27.5}{4.78} \approx -2.61$$

$$\Phi^{-1}(pval) = -2.61, \;\; pval = 0.0045 \leq 0.05$$

In practice, these were computed with the Python Scipy library as follows:

```
import scipy.stats as stats
vib_scores = [0.915, 0.915, 0.91, 0.92, 0.89]
svib_scores = [0.93, 0.935, 0.925, 0.93, 0.94]
pvalue = stats.ranksums(svib_scores, vib_scores, 'greater').pvalue
assert pvalue < 0.05
```

### IMAGE CLASSIFICATION

The ImageNet 2012 validation set was used for evaluation as the test set for ImageNet is unavailable. InceptionV3 yields a slightly worse single shot accuracy than inceptionV2 (80.4%) when run in a single model and single crop setting, however we've used InceptionV3 over V2 for simplicity. Each model was trained for 100 epochs. The entire validation set was used to measure accuracy and robustness to FGS attacks, while only 1% of it was used for CW attacks, as they are computationally expensive. Complete results are available in Section 5. Examples of successful attacks are shown in Figures 6,7. t-SNE (van der Maaten & Hinton 2008) visualization of the latent space of each model is presented in Figure 8.

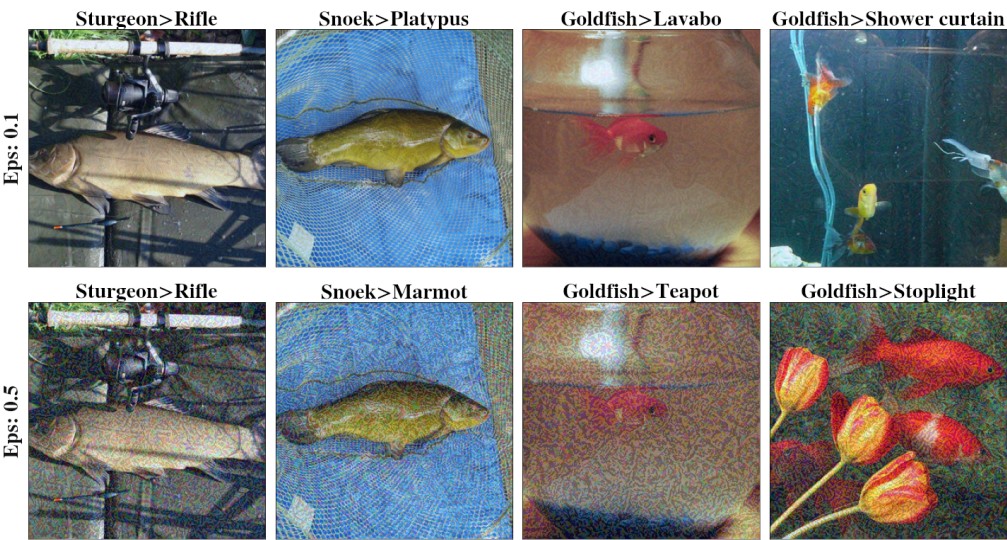

Figure 6: Successful untargeted FGS attack examples. Images are perturbations of previously successfully classified instances from the ImageNet validation set. Perturbation magnitude is determined by the parameter $\epsilon$ shown on the left, the higher, the more perturbed. Original and wrongly assigned labels are listed at the top of each image. Notice the deterioration of image quality as $\epsilon$ increases.

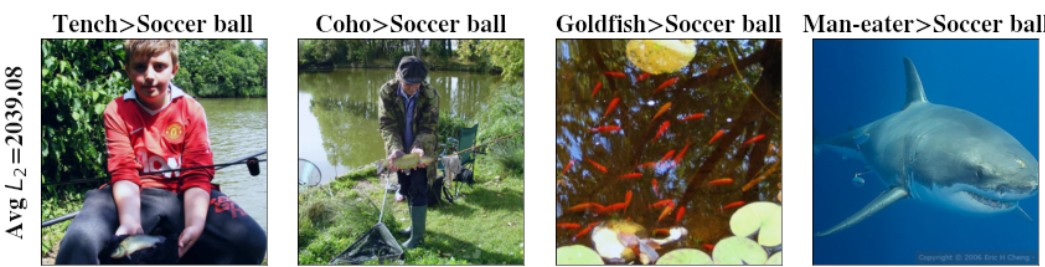

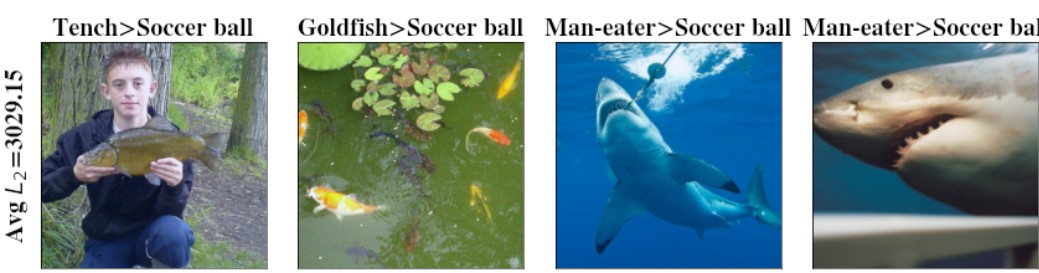

Figure 7: Successful targeted CW attack examples. Images are perturbations of previously successfully classified instances from the ImageNet validation set. The target label is 'Soccer ball'. Average $L_2$ distance required for a successful attack is shown on the left. The higher the required $L_2$ distance, the greater the visible change required to fool the model. Original and wrongly assigned labels are listed at the top of each image. Mind the difference in noticeable change as compared to the FGS perturbations presented in Figure 6.

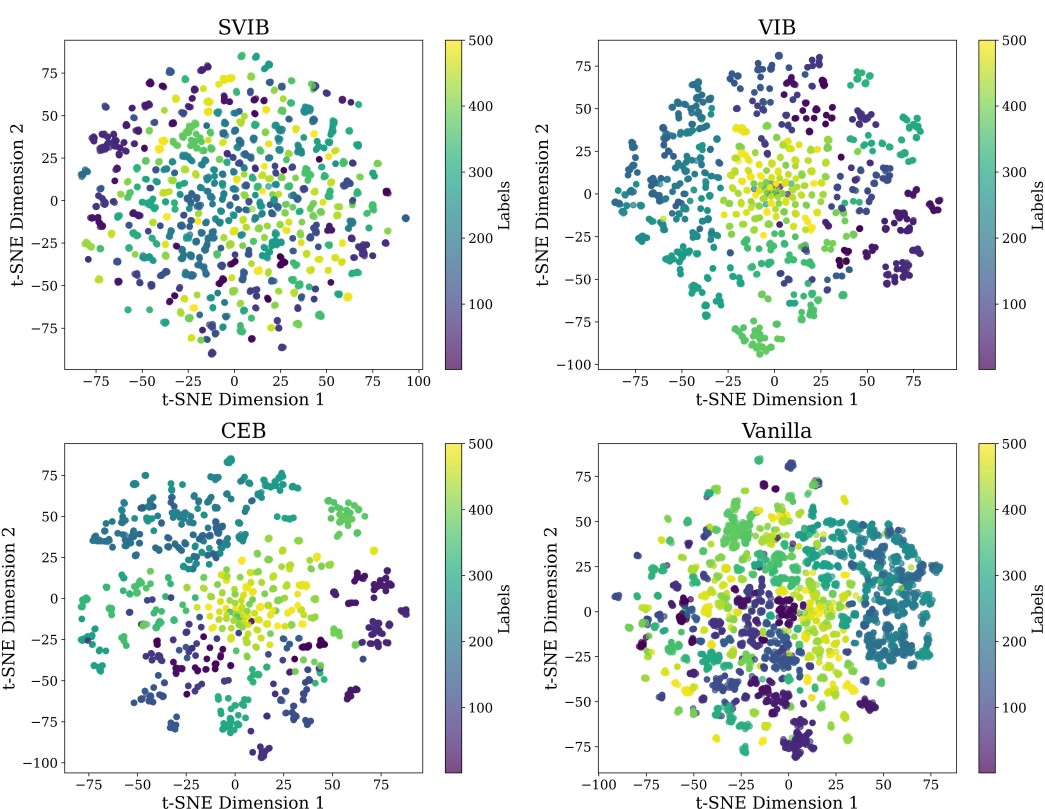

Figure 8: ImageNet embeddings of the different models casted to 2D using the t-SNE algorithm (van der Maaten & Hinton 2008). 5000 datapoints of the first 500 ImageNet labels. The VIB and CEB castings share similar traits of well separated clusters, while the vanilla casting shows some clustering that that seems less formed and unseparated. The SVIB casting shows very little clustering and features the most dispersed distribution. The visualization suggests that the conditional entropy term in SVIB has negated the clustering effect of the ELBO loss, and induced a more uniform representation.

TEXT CLASSIFICATION

Each model was trained for 150 epochs. The entire test set was used to measure accuracy, while only the first 200 entries in the test set were used for adversarial attacks, as they are computationally expensive. Complete results are available in Section 5. Examples of successful attacks are shown in Tables 1,2.

| **Original text** |
|---|
| the acting , costumes , music , cinematography and sound are all *astounding* given the production's austere locales. |
| **Perturbed text** |
| the acting , costumes , music , cinematography and sound are all *dumbfounding* given the production's austere locales. |

Table 1: Example of a successful PWWS attack on a vanilla Bert model, fine tuned over the IMDB dataset. The original label is 'Positive sentiment'. The substituted word, marked in italic font, changed the classification to 'Negative sentiment'. SVIB and VIB classifiers are far less susceptible to these perturbations as shown in Figure 2.

| **Original text** |
|---|
| *great* historical movie, will not allow a viewer to leave once you begin to watch. View is presented differently than displayed by most school books on this *subject*. My only fault for this movie is it was photographed in black and white; wished it had been in color ... wow ! |
| **Perturbed text** |
| *gnreat* historical movie, will not allow a viewer to leave once you begin to watch. View is presented differently than displayed by most school books on this *sSbject*. My only fault for this movie is it was photographed in black and white; wished it had been in color ... wow ! |

Table 2: Example of a successful Deep Word Bug attack on a vanilla Bert model, fine tuned over the IMDB dataset. The original label is 'Positive sentiment'. Perturbations, marked in italic font, change the classification to 'Negative sentiment'. SVIB and VIB classifiers are far less susceptible to these perturbations, as shown in Figure 2.

Complete empirical results

The following tables contain the results of all experiments run in this study.

| $\beta$ | $\lambda$ | Val $\uparrow$ | FGS $\uparrow$ $\epsilon=0.1$ | FGS $\uparrow$ $\epsilon=0.5$ | CW$\uparrow$ |
|---|---|---|---|---|---|
| **Vanilla model** | | | | | |
| - | - | 77.2% | 31.1% | 32.3% | 788 |
| **SVIB models** | | | | | |
| $10^{-4}$ | 2 | 75.4% $\pm.01\%$ | 40.1% $\pm.08\%$ | 33.7% $\pm2.1\%$ | 3401 $\pm267$ |
| $10^{-3}$ | 0.5 | 74.9% $\pm.06\%$ | 38.4% $\pm.06\%$ | 33.8% $\pm.1\%$ | 3293 $\pm140$ |
| $10^{-3}$ | 1 | 75.5% $\pm.03\%$ | 37.2% $\pm.1\%$ | 33.6% $\pm.1\%$ | 2666 $\pm140$ |
| $10^{-3}$ | 2.0 | 75.4% $\pm.07\%$ | 38.1% $\pm.1\%$ | 33.7% $\pm.1\%$ | 2981 $\pm260$ |
| $10^{-3}$ | 2.5 | 75.3% $\pm.01\%$ | 38.3% $\pm.2\%$ | 33.8% $\pm.15\%$ | 3095 $\pm407$ |
| $10^{-3}$ | 3.0 | 75.3% $\pm.03\%$ | 38.5% $\pm.2\%$ | 33.9% $\pm.16\%$ | 3078 $\pm443$ |
| $10^{-2}$ | 0.5 | 74.2% $\pm.11\%$ | 42.0% $\pm.13\%$ | 35.2% $\pm.06\%$ | 2354 $\pm394$ |
| $10^{-2}$ | 1 | 75.0% $\pm.05\%$ | 42.4% $\pm.2\%$ | 35.7% $\pm.1\%$ | 1564 $\pm218$ |
| $10^{-2}$ | 2.0 | 75.3% $\pm.07\%$ | 43.1% $\pm.1\%$ | 36.3% $\pm.1\%$ | 1748 $\pm160$ |
| $10^{-2}$ | 2.5 | 75.4% $\pm.06\%$ | 43.0% $\pm.13\%$ | 36.0% $\pm.1\%$ | 1814 $\pm144$ |
| $10^{-2}$ | 3.0 | 75.4% $\pm.07\%$ | 42.9% $\pm.18\%$ | 36.2% $\pm.12\%$ | 1749 $\pm138$ |
| $10^{-1}$ | 0.5 | 73.1% $\pm.04\%$ | 39.1% $\pm.2\%$ | 32.6% $\pm.19\%$ | 3738 $\pm138$ |
| $10^{-1}$ | 1 | 74.8% $\pm.09\%$ | 42.1% $\pm.5\%$ | 35.2% $\pm.5\%$ | 3575 $\pm456$ |
| $10^{-1}$ | 2.0 | 75.4% $\pm.03\%$ | 46.6% $\pm1.8\%$ | 39.8% $\pm2.1\%$ | 3332 $\pm443$ |
| $10^{-1}$ | 2.5 | 75.4% $\pm.03\%$ | 45.6% $\pm1.2\%$ | 38.7% $\pm1.3\%$ | 3581 $\pm243$ |
| $10^{-1}$ | 3.0 | 75.1% $\pm.09\%$ | 46.0% $\pm.8\%$ | 39.3% $\pm1.0\%$ | 3536 $\pm315$ |

Table 3: Complete ImageNet evaluation scores for vanilla and SVIB models, average over 5 runs with standard deviation. First column is performance on the ImageNet validation set, second and third columns are the percent of unsuccessful FGS attacks at $\epsilon = 0.1, 0.5$, and the fourth column is the average $L_2$ distance for a successful Carlini Wagner $L_2$ targeted attack. For all columns higher is better $\uparrow$.

| $\beta$ | $\rho$ | Val $\uparrow$ | FGS $\uparrow$ $\epsilon=0.1$ | FGS $\uparrow$ $\epsilon=0.5$ | CW$\uparrow$ |
|---|---|---|---|---|---|
| **VIB models** | | | | | |
| $10^{-4}$ | - | 74.8% $\pm.01\%$ | 28.3% $\pm.2\%$ | 29.3% $\pm.2\%$ | 1554 $\pm280$ |
| $5 \cdot 10^{-4}$ | - | 74.1% $\pm.01\%$ | 37.7% $\pm.01\%$ | 34.8% $\pm.01\%$ | 3104 $\pm529$ |
| $10^{-3}$ | - | 73.7% $\pm.1\%$ | 40.5% $\pm.2\%$ | 36.1% $\pm.2\%$ | 3917 $\pm291$ |
| $5 \cdot 10^{-3}$ | - | 73.0% $\pm.04\%$ | 44.9% $\pm.13\%$ | 37.8% $\pm.21\%$ | 3358 $\pm245$ |
| $10^{-2}$ | - | 72.8% $\pm.1\%$ | 46.5% $\pm.2\%$ | 38.0% $\pm.1\%$ | 3318 $\pm293$ |
| $5 \cdot 10^{-2}$ | - | 72.3% $\pm.07\%$ | 44.7% $\pm.3\%$ | 34.9% $\pm.32\%$ | 3654 $\pm333$ |
| $10^{-1}$ | - | 72.1% $\pm.01\%$ | 41.6% $\pm.1\%$ | 38.0% $\pm.1\%$ | 3318 $\pm293$ |
| $5 \cdot 10^{-1}$ | - | 0.1% $\pm0\%$ | 0% $\pm0\%$ | 0% $\pm0\%$ | 0 $\pm0$ |
| **CEB models** | | | | | |
| - | 1 | 73.0% $\pm.07\%$ | 26.5% $\pm.22\%$ | 28.7% $\pm.15\%$ | 4527 $\pm64$ |
| - | 2 | 73.2% $\pm0\%$ | 26.4% $\pm.21\%$ | 29.0% $\pm.03\%$ | 4342 $\pm173$ |
| - | 3 | 73.4% $\pm0\%$ | 26.7% $\pm.12\%$ | 29.3% $\pm.18\%$ | 4556 $\pm177$ |
| - | 4 | 73.8% $\pm.08\%$ | 27.0% $\pm0\%$ | 29.9% $\pm.07\%$ | 3689 $\pm347$ |
| - | 5 | 74.3% $\pm.05\%$ | 27.6% $\pm.13\%$ | 30.1% $\pm.22\%$ | 1776 $\pm146$ |
| - | 6 | 74.6% $\pm.03\%$ | 27.7% $\pm.35\%$ | 30.0% $\pm.13\%$ | 1103 $\pm154$ |
| - | 7 | 74.6% $\pm.04\%$ | 28.0% $\pm.02\%$ | 30.1% $\pm.02\%$ | 847 $\pm16$ |

Table 4: Complete ImageNet evaluation scores for VIB and CEB models, average over 5 runs with standard deviation. First column is performance on the ImageNet validation set, second and third columns are the percent of unsuccessful FGS attacks at $\epsilon = 0.1, 0.5$, and the fourth column is the average $L_2$ distance for a successful Carlini Wagner $L_2$ targeted attack. For all columns higher is better $\uparrow$.

| $\beta$ | $\lambda$ | Test↑ | DWB↑ | PWWS↑ |
|---------|-----------|-------|------|-------|
| **Vanilla model** | | | | |
| - | - | 93.0% | 45.7% | 0.0% |
| **SVIB models** | | | | |
| $10^{-4}$ | 1 | 92.4% ±.01% | 68.4% ±1.7% | 63.9% ±3.3% |
| $10^{-3}$ | 0.5 | 92.3% ±.07% | 70.7% ±2.3% | 68.3% ±3.3% |
| $10^{-3}$ | 1 | 93.2% ±.5% | 72.5% ±2.0% | 71.6% ±1.3% |
| $10^{-3}$ | 2.0 | 92.3% ±.07% | 74.7% ±3.5% | 73.1% ±3.4% |
| $10^{-3}$ | 2.5 | 92.4% ±.07% | 75.9% ±1.9% | 72.4% ±1.8% |
| $10^{-3}$ | 3.0 | 92.3% ±.04% | 74.5% ±1.7% | 74.4% ±.9% |
| $10^{-2}$ | 0.5 | 92.4% ±.06% | 66.1% ±4.2% | 68.3% ±3.3% |
| $10^{-2}$ | 1 | 92.6% ±.8% | 69.2% ±2.0% | 50.0% ±4.8% |
| $10^{-2}$ | 2.0 | 92.4% ±.1% | 64.8% ±4.7% | 40.3% ±7.4% |
| $10^{-2}$ | 2.5 | 92.3% ±.1% | 58.1% ±2.5% | 28.9% ±2.45% |
| $10^{-2}$ | 3.0 | 92.3% ±0.1% | 54.0% ±3.3% | 22.5% ±2.6% |
| $10^{-1}$ | 0.5 | 92.2% ±0.02% | 1.1% ±1.1% | 0.0% ±0% |
| $10^{-1}$ | 1 | 89.2% ±2.0% | 0.8% ±0.5% | 0.0% ±0% |
| $10^{-1}$ | 2.0 | 92.3% ±.2% | 0.0% ±0% | 0.0% ±0% |
| $10^{-1}$ | 2.5 | 92.4% ±.1% | 0.0% ±0% | 0.0% ±0% |
| $10^{-1}$ | 3.0 | 92.4% ±.1% | 0.0% ±0% | 0.0% ±0% |

Table 5: Complete IMDB evaluation scores for vanilla and SVIB models, average over 5 runs with standard deviation. First column is performance over the test set, second is percent of unsuccessful Deep Word Bug attacks, and third column is percent of unsuccessful PWWS attacks. For all columns higher is better ↑.

| $\beta$ | $\rho$ | Test↑ | DWB↑ | PWWS↑ |
|---|---|---|---|---|
| **VIB models** | | | | |
| $10^{-4}$ | - | 92.1% ±1.1% | 67.0% ±3.2% | 60.8% ±1.4% |
| $5 \cdot 10^{-4}$ | - | 92.2% ±.07% | 68.2% ±3.0% | 64.3% ±1.3% |
| $10^{-3}$ | - | 91.0% ±1.0% | 64.9% ±4.4% | 58.4% ±6.6% |
| $5 \cdot 10^{-3}$ | - | 92.2% ±.07% | 62.9% ±3.9% | 48.3% ±7.5% |
| $10^{-2}$ | - | 90.8% ±0.5% | 59.0% ±4.8% | 37.1% ±14.3% |
| $5 \cdot 10^{-2}$ | - | 92.4% ±.1% | 14.4% ±5.5% | 1.0% ±0.3% |
| $10^{-1}$ | - | 89.4% ±.9% | 10.0% ±8.0% | 0.9% ±0.9% |
| **CEB models** | | | | |
| - | 0.1 | 92.7% ±.04% | 46.7% ±0.68% | 1.65% ±0.27% |
| - | 1 | 92.7% ±0% | 43.2% ±1.45% | 1.53% ±0.8% |
| - | 2 | 92.5% ±0% | 40.8% ±.72% | 0% ±0% |
| - | 3 | 92.3% ±0% | 36.2% ±0% | 0% ±0% |
| - | 4 | 92.1% ±0% | 38.8% ±0% | 0% ±0% |
| - | 5 | 92.2% ±0% | 39.6% ±0% | 1.0% ±0% |
| - | 6 | 92.1% ±0% | 41.9% ±0% | 0% ±0% |
| - | 7 | 92.2% ±0% | 41.9% ±0% | 2.15% ±0% |
| - | 8 | 92.2% ±0% | 45.9% ±0% | 0% ±0% |

Table 6: Complete IMDB evaluation scores for VIB and CEB models, average over 5 runs with standard deviation. First column is performance over the test set, second is percent of unsuccessful Deep Word Bug attacks, and third column is percent of unsuccessful PWWS attacks. For all columns higher is better ↑.

