# OpenReview forum: "Revisiting the Variational Information Bottleneck"
_ICLR.cc/2025/Conference — Submitted to ICLR 2025_

### Official Review · Reviewer_jzXn · 2024-10-31

**Soundness:** 3
**Presentation:** 3
**Contribution:** 2
**Rating:** 6
**Confidence:** 4

**Summary:**

The paper extends the variational information bottleneck by adding an entropy regularizer to the model that predicts the target y given the latent z. This is motivated by adding and variational bounding a second info bottleneck.

**Strengths:**

The paper boils down to a simple to implement and intuitive loss function.

**Weaknesses:**

There is a lot of justification that is somewhat verbose and subjective. The derivation is long and elaborate for what boils down to an extra regularizer with an extra tuning parameter.

**Questions:**

Here are some detailed comments and questions:

Use \log in latex and format the integral d.

The writing may be a little verbose. Examples: lines 260-264 restates things. (7) follows trivially from (4). The sentences preceding both (4) and (7) are also similar. lines 217-254 is repeat well known material.

Line 198 why carry p(x,y) around if everything is conditional on those later? I suspect dropping that saves some hassle later.

Lines 271 to 296 would be better expanded after moving lines 217-254 to an appendix. You could be explicit about the use of the chain factorization etc (although maybe the previous point about line 198 can avoid needing to deal with this?).

Line 249 who’s -> whose

Line 334 why is Z left as an r.v.? Please explain how to handle this with sampling. I feel like this is just the entropy of the y given the sampled z, so it should be written explicitly as such?

Line 452 Val column bolded wrongly, the vanilla method should be shown as the winner not the proposed method?

Table 1: it seems as though VIB best performance is at the boundary of your sweep, so we can’t tell if SVIB beats VIB?

Table 2: as previous comment.

A plot instead of tables 5 and 6 would be easier to absorb.

Tables 1 and 2: can you not fix lambda and show improvement generally? Varying this in-sample looks like overfitting to the untrained eye (but I think this is an illusion and the results are good). It just seems like sub optimal presentation given tables 5 and 6 show good robust performance over lambda. A plot >> tables of numbers.

Section 4: showing robustness is nice, and the methodology seems very good i.e. adversarial approaches.

Line 478 private -> special

---

> ### Author Response · Authors · 2024-11-26
> **Rebuttal response**
>
> We thank the reviewer for their meaningful, detailed and valuable feedback. We are pleased the reviewer found our proposed loss intuitive and easy to implement, and our experimental methodology good.
>
> We note that we've uploaded a revised version of our submission, the details of which are specified in the general comment.
>
> **The reviewer notes places where writing is too verbose**
>
> We thank the reviewer for their detailed remarks. We note that we've actively chose verbosity in an attempt to make the paper more accessible to scholars that are unfamiliar with IB. We address the individual comments as follows:
>
> $\underline{\text{Lines 260-264}}$
>
> We've amended this paragraph to be shorter.
>
> $\underline{\text{Eq (7) follows trivially from (4), and the sentences preceding Eq (4) Eq (7) are similar}}$
>
> We've shortened the derivation of Eq (8) by directly citing the Barber-Agakov bound [1] instead of deriving Eq (7), which we've discarded. This change also solves the restatement of the non negativity of KL divergence.
>
> $\underline{\text{Lines 217-254 repeats previous work}}$
>
> The reviewer is correct, and we've considered whether to move this derivation to an appendix. We chose to leave it in the body because the clause in lines 248-253 is an important statement that links the two bottlenecks together. Without setting $\tilde{Y}|Z=z~c(\cdot|z)$, and without establishing that the VIB inequality holds for any conditional distribution $c(\cdot|z)$, the second bottleneck can reach a trivial solution where $I(Z|\tilde{Y})=0$. We believe it would be harder to convey this message when the derivation of the original VIB bound is moved to an appendix.
>
> **The reviewer notes we should consider not optimizing over $p(x,y)$ when proposing the new bottleneck in line 198**
>
> We've added $p(x,y)$ as an argument to Eq (2) to clarify the clause above, in particular lines 188-190. We accept the reviewer suggestion to remove it.
>
> **The reviewer notes that lines 271-296 should be better expanded**
>
> Follows from the last note.
>
> **The reviewer asks why $Z$ is left as a RV in line 334**
>
> We wrote the regularization term as explicit entropy in an attempt to make Eq (16) intuitive, which resulted in notation abuse. We've fixed this in the revised submission by defining the instance $\tilde{y}_{n}$, and changed the term to:
>
> $$\lambda \log\left(C_{\gamma}(\tilde{y}_n \mid \hat{z}_n)\right)$$
>
> We thank the reviewer for pointing this out.
>
> **The reviewer notes that SVIB only slightly outperforms VIB**
>
> Experiments show that SVIB outperforms VIB in terms of accuracy by up to 3.3% absolute, while achieving higher robustness to all attacks expect for the CW attack. It is true for the FGS attacks SVIB only slightly outperforms VIB, but we believe the trend is clear enough, especially when comparing the best VIB and SVIB models. We've improved the visualization of the results by using plots as suggested, and we've added experiments for VCEB further substantiate SVIB's performance gains.
>
> **The reviewer suggests using plots instead of tables**
>
> We've changed tables 1,2 to plots with a fixed $\lambda$ value as required, and chose to keep tables 5,6 as tables. We find this new presentation a big improvement in readability, and we thank the reviewer for suggesting this.
>
> We note that we've also added many new VIB experiments, stretching the range of relevant $\beta$ values, balancing the amount of experiments between methods.
>
> **The reviewer notes a wrongly bolded value in line 452**
>
> Boldface amended.
>
> **The reviewer notes the proper usage of Latex and grammar errors**
>
> We've amended the errors as required, we thank the reviewer for pointing these out.
>
> **References**
>
> [1] Barber, D.; and Agakov, F. V. 2003. The IM algorithm: a variational approach to Information Maximization. In Neural Information Processing Systems.
>
> [2] The Conditional Entropy Bottleneck, Entropy 2020.

---

### Official Review · Reviewer_kcy5 · 2024-10-31

**Soundness:** 2
**Presentation:** 3
**Contribution:** 3
**Rating:** 5
**Confidence:** 5

**Summary:**

- The paper picks up an issue identified in the Deep Variational Information Bottleneck paper, where the classifier can overfit to the learned representation, Z, of a VIB model, and proposes a new framework for supervised learning with IB, which they call Supervised Information Bottleneck (SIB), and a corresponding variational approach, SVIB.
- The core theoretical contribution is to add a constraint to the IB and VIB objectives that minimizes an upper bound on I(\hat{Y},Z), which is equivalent to maximizing a lower bound on H(\hat{Y}|Z). The paper shows that this new constraint is tractable in the SVIB setting.
- The paper provides experiments comparing SVIB to VIB and “vanilla” Maximum Likelihood models (trained with cross entropy) on ImageNet and natural language sentiment analysis.

**Strengths:**

- The paper is well-written and easy to read.
- Constrained maximization of H(\hat{Y}|Z) will clearly achieve the goal of preventing the classifier from overfitting to the representation.
- The theoretical approach is plausibly useful. A careful set of experiments could demonstrate its value beyond using VIB or CEB.

**Weaknesses:**

- In general, the experiments are of the correct form (comparisons between different IB approaches and Maximum Likelihood on clean and adversarial test sets), but they are unconvincing at supporting the main claim that SVIB substantially improves on other proposed tractable IB approaches, as pointed out in more detail below.
- One shortcoming of all of the experiments is that the VIB models are not given the same amount of hyperparameter tuning as the SVIB models – it appears that in all cases, the SVIB models get three times as many runs with different hyperparameters to find a setting that outperforms the VIB models.
- VIB on classification tasks often benefits from having a mixture distribution for r(z), whether learned or just distributed across part of the domain of Z, rather than having a single isotropic Gaussian distribution for r(z). It’s likely that your selected values of \beta would perform better in that setting, as it becomes easier for the model to learn to assign classes to different mixture elements as it sees fit, which makes the model more powerful (more powerful models can tolerate higher compression/higher values of \beta). This would likely benefit SVIB as well, so that it more reliably outperforms the Maximum Likelihood baseline on the test set.
- The paper is missing an important citation: CEB Improves Model Robustness, Entropy 2020. Overlooking this reference is a major shortcoming of the paper, since it studies the same question on one of the same datasets using the same Information Bottleneck framework, and it achieves substantially better results on that dataset than reported in this paper (its VIB results are also stronger than your VIB and SVIB results).
- The ImageNet table highlights SVIB results in settings where the VIB results appear to strongly overlap – it seems a stretch to claim that SVIB is doing better than VIB with a result of 53.4%+/-1.8% compared to 53.5%+/-0.2% for FGS with \epsilon=0.1, for example (and similarly but to a lesser extent for FGS with \epsilon=0.5).
- For all experiments, hyperparameter selection for VIB is questionable, as test set performance on the clean data appears to still be improving substantially at the smallest value of \beta. As \beta goes to 0, its performance should match the vanilla model on the clean data, but you stop exploring \beta when the test set performance is substantially worse than the vanilla model, indicating that probably neither the VIB nor the SVIB models are very close to optimally configured.
- The Conditional Entropy Bottleneck paper showed that CEB reliably outperforms VIB on both clean and adversarial examples on a variety of image datasets. The CEB Improves Model Robustness paper further explores that in detail on ImageNet. Since implementing CEB can be made parameter-equivalent to implementing VIB (and consequently SVIB), it seems like an important point of comparison.
- In Figure 1, right-hand side, the H(\hat{Y}) circle is drawn in a way that does not respect the Markov chain constraint Y-X-Z-\hat{Y}. It is not possible to have H(\hat{Y}) overlap H(Y) in any area where H(Z) does not also overlap H(Y). Compare this to the Venn diagrams in the Conditional Entropy Bottleneck paper you cite, where similarly the Markov chain Z-X-Y prevents H(Z) from overlapping H(Y) anywhere that H(X) does not also overlap H(Y).
- Line 360: repeated word: “is uninformative about about Y”.

**Questions:**

- I think the theoretical contribution is solid and valuable to share with the community, but I think the empirical treatment is weak. I would be very happy to increase my rating if the experiments were improved, even if they did not show that SVIB is reliably better than VIB or CEB in all of the settings considered. Whatever the outcome for SVIB on more careful preliminary experiments would be a valuable scientific contribution.

---

> ### Author Response · Authors · 2024-11-26
> **Rebuttal response**
>
> We thank the reviewer for their meaningful, detailed and valuable feedback. We note that it's assuring to read that the reviewer found our paper easy to read, and our contribution valuable.
>
> We note that we've uploaded a revised version of our submission, the details of which are specified in the general comment.
>
> **The reviewer mentions an important work in the field: CEB Improves Model Robustness [1]**
>
> We thank the reviewer for pointing out this important work that we we're not aware of. We've added extensive CEB experiments to the revised paper, took a deeper dive into CEB and [1] in the related work section, and added the CEB implementation to our submitted code.
>
> As elaborated in the revised submission, CEB outperformed SVIB on the CW targeted attack, and outperformed VIB in test accuracy for both modalities, and SVIB outperformed CEB and VIB in all other tasks. These results are on par with [1], where it was shown that a ResNet50 model outperformed vanilla ResNet50 on the targeted PGD attack, and on accuracy scores for adversarial flavors of ImageNet. We note that ResNet50 is similar in parameter size to the Inception V3 model used in our experimental setting.
>
> We note the differences in the experimental setting in [1] and our work, which is a replication and extension of the setting in the original VIB [4] paper: Most notably, [1] uses ImageNet-A and ImageNet-C, while we use vanilla ImageNet. [1] mostly tests larger sized DNNs of ~60M parameters, while we only use Inception which is ~24M parameters. Moreover, [1] tests a single attack for ImageNet (targeted PGD), runs much larger batch sizes, does not use frozen pre-trained encoders, and does not compare to VIB on ImageNet.
>
> **The reviewer notes that VIB and SVIB results closely overlap**
>
> Experiments show that SVIB outperforms VIB in terms of accuracy by up to 3.3% absolute, while achieving higher robustness to all attacks expect for the CW attack. While SVIB only slightly outperforms VIB on the FGS attack, we believe the trend is consistent, especially when comparing the best VIB and SVIB models. We've improved the visualization of the results by using plots in the revised paper, and we note that the experiments for VCEB further substantiate SVIB's performance gains.
>
> All in all, we feel that these results are sufficient to ground out proposed method, and provide an easy to implement improvement in accuracy and robustness to VIB based models.
>
> **The reviewer notes that hyperparameter search for VIB is lacking**
>
> We thank the reviewer for pointing out this discrepancy. We haven't considered the effects of multiple testing when devising our experiments, as we should have. We've extended the VIB experiments in the new submission to a wider range of beta values. VIB experiments now cover all values from 0.5 to $10^{-4}$, such that we've stopped exploring new values once performance was clearly diminished. While SVIB still has more experiments than VIB, we note that we've covered all ranges of useful beta values for VIB, and that SVIB shows consistent results per $\beta$ value for the different $\lambda$ combinations.
>
> **The reviewer notes that VIB can significantly benefit by using GMM distribution**
>
> We thank the reviewer for pointing this out. Indeed, we accept that a stronger model could tolerate higher regularization for both $\beta$ and $\gamma$. We've added this important remark to the future work clause, together with a CEB-SVIB combination. We note that it is exciting to see what these combinations, together with CEB rate terms, can achieve.
>
> **The reviewer notes that Figure 1 does not adhere to the Markov chain $Y-X-Z-\tilde{Y}$**
>
> We've amended the Figure as required. Once again, we thank the reviewer for this important note.
>
> **References**
>
> [1] CEB Improves Model Robustness, Entropy 2020.
>
> [2] The Conditional Entropy Bottleneck, Entropy 2020.
>
> [3] The Information Bottleneck Method. The 37th annual Allerton Conference 1999
>
> [4] Deep Variational Information 260 Bottleneck. ICLR 2017
>
> [5] A Comparison of Variational Bounds for the Information Bottleneck Functional. Entropy 2020

---

> > ### Comment · Reviewer_kcy5 · 2024-11-27
> >
> > Thank you for the detailed response, and for the additional experimental results.
> >
> > **Most notably, [1] uses ImageNet-A and ImageNet-C, while we use vanilla ImageNet...**
> > There seems to be a misunderstanding here -- ImageNet-A and ImageNet-C are additional test sets for models trained on ImageNet. They are not meant to be used during training. The ImageNet models in [1] are all trained on vanilla ImageNet, not using ImageNet-A or ImageNet-C.
> >
> > **Moreover, [1] tests a single attack for ImageNet (targeted PGD)**
> > You are probably aware of this, but targeted PGD is a much more challenging attack than FGS (FGS is a single step of either targeted or untargeted PGD, depending on the attack loss you use). [2] also shows strong results on CW for both VIB and CEB, although that work does not study ImageNet.
> >
> > The strength of the results in [1] and [2] on a variety of different adversaries makes me wonder whether your implementations of VIB and CEB are correct, since you are showing both poor accuracy on the validation sets and poor accuracy on the adversarial examples for VIB and CEB. If the differences in results between those previous works and your work are not due to implementation bugs, my guess is that your models are weak. Both the smaller Inception model and the apparent lack of learned parameters for the marginal m(z) are substantial weaknesses compared to the experiments in [1] and [2]. It isn't clear to me why these weaknesses are appropriate, given that the literature clearly shows how to make more powerful VIB and CEB models than what was proposed in [4], which is 9 years old at this point and therefore not a strong choice for the experimental setting. In fact, I understand now that you are following the ImageNet experimental design in [4], which took a pretrained ImageNet model and replaced the final linear layer with a VIB classifier layer trained on the fixed ImageNet representation. That was due to the expense at the time of training an ImageNet model from scratch. The other experiments in [4] did not do that, as it is highly suboptimal – the VIB (and CEB) loss should give gradients to the entire model throughout training if at all possible. Since ImageNet models are very cheap and fast to train now, I don’t see a reason to do the experiments in this manner.
> >
> > Also, the results in [1] and [2] fairly clearly indicate that weaker models perform better when beta is smaller or rho is larger. I noticed that the range of rho for the CEB models only goes up to 7 in your revisions, whereas the much more powerful models in [1] had optimal validation performance at rho=6 for the weakest model (so rho=7 in your setting is unlikely to be sufficient for optimal validation performance), and explored up to rho=10.
> >
> > This raises another concern with the experimental setup -- since we can't know in advance what adversaries are going to be used, it is not reasonable to select hyperparameters based on performance on adversarial examples. [1] resolves this by selecting the model based on the non-adversarial validation set performance, and then studies how well those models perform in the adversarial settings. [1] clearly demonstrated that the models can be both more accurate on clean validation set examples and on unknown adversaries.
> >
> > While I appreciate the effort the authors have put into responding to my concerns and improving the paper, I am still uncomfortable with the quality of the experimental results. I am prepared to believe that SVIB does outperform VIB and CEB under fair comparisons, and I think that would be a very valuable contribution to the community. However, as it currently stands, the empirical evidence seems too questionable, considering how much better those two approaches performed in the previous literature compared to how they are presented here. I would suggest that the authors perform more careful experiments that replicate the previously reported performance of VIB and CEB in order to make the benefits of SVIB unimpeachably clear; otherwise this potentially important work may be largely overlooked. That level of reworking of the experiments is certainly too much to ask for in revision, so I am not comfortable recommending the paper for acceptance at this time.

---

> ### Author Response · Authors · 2024-11-27
>
> We thank the reviewer for their response, and are encouraged to read that they see our work as potentially important.
>
> The reviewer's main concern is about the quality of our experiments, most notably that our VIB and CEB implementation seem weaker than previous work in terms of accuracy and robustness.
>
> **Comparison of tests performed**
>
> - In [2] CEB demonstrated improved accuracy and robustness to targeted PGD attack for F-MNist, and increased robustness to untargeted PGD on Cifar-10, compared to VIB and Vanilla.
>
> - In [1] CEB demonstrated improved accuracy and robustness to untargeted PGD on Cifar-10, and improved robustness to targeted PGD for ImageNet, compared to Vanilla.
>
> - In [4] VIB demonstrated slight decrease in accuracy over ImageNet test set, and an increase in robustness to untargeted FGS and targeted CW attacks, compared to vanilla model.
>
> - Our experiments compare CEB, VIB, Vanilla, and our proposed SVIB on ImageNet and IMDB dataset, over 3 untargeted attacks, and the targeted CW attack. We show that CEB yields a dramatic increase in robustness to the targeted CW attack, and better test set accuracy than VIB. With regards to the untargeted attacks, since [1] and [2] didn't report results for untargeted attacks for ImageNet (as they did for Cifar), and did not compare CEB to VIB on ImageNet, there really is no contradiction.
>
> - With regards to VIB, our experiments are on par with [4], showing the same slight decrease in test set accuracy, and the same sharp increase in robustness to both attacks.
>
> Given the challenges of extrapolating results from CIFAR-10 to ImageNet, it is worth considering why prior work reported untargeted attacks only on CIFAR-10 and not on ImageNet, and why a comparison of CEB to VIB on ImageNet was not included, despite CEB being a direct improvement. By incorporating additional modalities and attacks, our work extends and complements prior efforts, without contradicting them.
>
> **Flaws in our experiments**
>
> We ask to restate that the goal of our experiments is not to benchmark the best performance globally. Rather it is to ground our theoretical work, which is the main contribution of the paper. We ask to cite this sentence from the experimental section of [2]: "Our experiments focus on comparing the performance of otherwise identical models when we change only the objective function and vary 𝜌. Thus, we are interested in relative differences in performance that can be directly attributed to the difference in objective and 𝜌.".
>
> On the other hand, [1] is a thorough empirical study, and its main contribution is to rigorously test CEB, which was proposed in [2]. Indeed, [1] applies many more methods to train better models, such as testing 3 different architectures, pretraining with AutoAug, testing 2 different types of CEB classifiers, L_2 weight decay, rho-annealing and more. All together training 86 ImageNet models from the ground up.
>
> We couldn't compete with this scale of testing. We devised our experiments to be as close as possible to the ones proposed in [3], as we propose a direct improvement over VIB. We've added a modality and 2 more attacks to further substantiate our findings, and we decided on 5 runs per setting to show statistical significance. Training this amount of ImageNet and IMDB models from the ground up would have been impossible for us from a resource perspective, so we've used pre-trained encoders of medium size Inception models.
>
> While our experiments are far from perfect, we firmly believe they are sufficient for the purpose of our claims, and in any case we cannot generate this scale of experimentation in the future.
>
> Following the reviewer's remark, we've added a qualifying sentence to our experimental section, resembling the one cited from [2]: "We note that our experiments compare identical models, varying only in objective functions and scaling parameters. This design highlights performance differences solely due to these factors. Methods like training from scratch to boost overall performance were omitted to ensure a robust comparison".
>
> In addition, we've reviewed our code following the reviewer's remark, and we ask to affirm that it sound. It is attached in the supplementary material for the reviewer's discretion.
>
> **Ranges of $\rho$ tested**
>
> From what we understood from [1], the best results for ResNet50 were for $\rho=3$, and ResNet50 is roughly the same parameter size as our Inception model. In addition, we've explored $\rho$ values of up to 10 (and also below 1), and they did not show improvement. These results were not reported because we only reported experiments that we've managed to run 5 times.
>
> **Selecting hyper parameters** I'm not sure I understood the reviewer's remark. We ran all attacks on all the models we've trained, and presented the averaged scores with std on the plot. Could you please explain this again?
>
> Thank you again for your time and insights.

---

> > ### Comment · Reviewer_kcy5 · 2024-11-30
> >
> > Thank you for the further explanation.
> >
> > **Selecting hyper parameters**
> >
> > My apologies for the lack of clarity in the concern I was raising. I think [1] did a good job of using a principled approach to model selection when presenting a method that improves robustness to adversarial attacks. In my understanding, the argument is something like the following. The person deploying a model doesn't know what adversaries the deployed model will face, so they shouldn't decide which model to select by looking at some set of adversaries and deploying the model that is best on those adversaries. Instead, they need some other decision criteria to select the model to deploy. Their preferred option, all else being equal, is to choose the model with the highest validation set accuracy on clean examples, as that model is expected to perform best for well-behaved users (users who aren't trying to attack the model, for example). Given a choice between models with comparable validation set performance, they should then prefer model families that have demonstrated robustness to a variety of attacks in the literature. In the case of [1] and [2], that corresponds to choosing the models with the highest compression (highest beta or lowest rho) that still achieve accuracy equal or better than the uncompressed (cross entropy) model.
> >
> > In contrast, in your presentation, the model selection question isn't addressed explicitly, unless I've overlooked it. Instead, it seems that implicitly you are suggesting that the person deploying a model should look at the adversarial attacks they think are interesting and choose the model that gets the best performance on those attacks. But those models generally have quite poor performance on clean examples, so the person deploying the model probably won't even consider deploying those models after reading your paper. This underscores the value of showing that IB approaches, well-implemented, are both better at clean examples and more robust to adversarial examples, as demonstrated in [1] and [2].
> >
> > **"We note that our experiments compare identical models, varying only in objective functions and scaling parameters. This design highlights performance differences solely due to these factors. Methods like training from scratch to boost overall performance were omitted to ensure a robust comparison"**
> >
> > This gets to the core of my concern about the experiments -- I believe that using a single pretrained representation harms the comparison, rather than making the comparison more robust. There is no reason to believe that the maximum likelihood (cross entropy) representation is the best for any of the IB approaches you consider, and there's no way to tell if that representation is biasing the results by being better for one of the approaches. Comparing the performance you show on clean validation examples versus what is reported on [1] and [2] makes it clear that this approach is harming at least VIB and CEB, since they aren't even close to matching the baseline performance on clean validation examples. I would guess that it's harming SVIB as well. But we don't know by how much, or if it's even the same amount for each approach. To me, this is a major flaw in the experiments.
> >
> > I appreciate that you feel that you don't have access to the resources necessary to train multiple models from scratch (and I'm not proposing that you need to do the level of empirical validation done in [1]). I'm not capable of judging your resource constraints. However, I would point out that ResNet50 ImageNet models can be trained from scratch in a matter of minutes on hardware from 2017 ([6] is an example of such a system). One of the strengths of VIB and CEB is that they converge in the same number of gradient steps as cross entropy models, so the VIB and CEB models can also be trained in minutes. I'm guessing the same is true for SVIB. I realize that your codebase may need modification to train that quickly. But given a codebase that trains that quickly, I think that a reasonable set of experiments on ImageNet could be done with a few hundred dollars of compute on the major cloud providers. This belief combined with the poor results in the current experiments compared with [1] and [2] is why I feel that it's not unreasonable to improve the experiments and resubmit. However, since I am not able to judge your resource constraints, and I don't want to unreasonably prevent your work from being accepted due to insurmountable resource constraints, I am happy to let the Area Chair evaluate whether my position on the experiments should block acceptance.
> >
> > [6] ImageNet Training in Minutes. https://arxiv.org/abs/1709.05011

---

> ### Author Response · Authors · 2024-12-04
>
> Thank you for your response and clarifications.
>
> **Selecting hyperparameters**
>
> The claim is that when comparing models against adversarial attacks, one must first pick the models according to a different criteria, in our case firstly test set accuracy, and then compression rate.
>
> Applying this evaluation to our experimental results we get:
>
> $\underline{\text{For IMDB}}$
>
> - SVIB models scored test accuracies of 92.4%, 93.2%, 92.6% and 89.2% for $\beta=10^{-4,-3,-2,-1}$ respectively. $10^{-3}$ has the highest accuracy by 0.6%, and achieves the best performance over all models in all settings and all metrics.
>
> $\underline{\text{For ImageNet}}$
>
> - SVIB models scored accuracies of 75.4%, 75.4%, 75.3% and 75.4% for $\beta=10^{-4,-3,-2,-1}$ respectively. $\beta=10^{-1}$ has the highest accuracy and compression, scoring 39.8% resilience to the FGS 0.5 attack and an $L_2$ distance of 3332 for the CW attack.
>
> - VIB models scored test accuracies of 74.8%, 74.1%, 73.7%, 73.0%, 72.8%, 72.3%, 72.1% and 0.1% for $\beta=10^{-4,-3.5,-3,-2.5,-2,-1.5,-1,-0.5}$ respectively. Making $\beta=10^{-4}$ the best setting in terms of accuracy, scoring 29.3% resilience to the FGS 0.5 attack and an $L_2$ distance of 1554 for the CW attack.
>
> - CEB models scored test accuracies of 74.6%, 74.6%, 74.3%, 73.8%, 73.4%, 73.2% and 73.0% for $\rho \in (7,6,...,1)$ respectively. $\rho=6$ has the highest accuracy, scoring 27.7% resilience to the FGS 0.5 attack and an $L_2$ distance of 1103 for the CW attack.
>
> We have that when comparing models chosen first by accuracy, and then by compression, the gap between the best SVIB and best VIB and CEB models is substantially bigger then when simply comparing their best performers task-wise.
>
> **Possible bias induced by MLE representations**
>
> The claim is that since our baseline test accuracy fell short of [1] and [2], it's possible that some bias is induced by the pretrained representations, hindering the validity of the experiments.
>
> We argue the following:
>
> - The baseline in [2], tested on Cifar-10 and FMNIST, is not directly comparable to our results on ImageNet and IMDB.
>
> - For IMDB we reach the vanilla accuracy both for VIB and SVIB.
>
> - Going from ~75% to 80% accuracy with ResNet50 on ImageNet depends mostly on hyperparameter tunning and various optimizations in training. This is a diminishing returns process, as is also evident in [6], where researchers reported being 1-3.4% behind the benchmark from ImageNet, due to lack of data augmentation.
>
> - Following the setting in [4], SVIB applies two ReLU activated FC layers before reparameterization, allowing feature extraction from the MLE representations.
>
> - While it is possible that a bias in favor of SVIB might occur, we don't see a reason to suspect this, apart from SVIB's improved performance. In that case, such a bias might exist between the different approaches given any change in input, such different modalities, which is not the case as shown by the consistent results on both ImageNet and IMDB.
>
> - Lastly, since [1] did not include a comparison of VIB models or untargeted attacks on ImageNet, we reiterate that our experimental results are consistent with theirs and with [4].
>
> **Resources required to run ground up training**
>
> Thank you for providing [6]. Data and model parallelism can speed up training by concentrating more resources at the same time, but won't reduce the aggregated amount of resources required. In our case, we have a quota of 64 vCPUs on AWS, allowing for a single p3.16xlarge, which is 25`$` an hour. Reproducing our experiments by training ImageNet from the ground up, and applying attacks, would take [7]: (8 VIB + 8 CEB + 12 SVIB) * (5 runs per setting) * (6 hours for ImageNet) * 25`$` = 21,000`$`.
>
> Training BERT from the ground up would take several days per setting [8].
>
> Reducing the experiments to ImageNet only, and 1 run per setting, would cost 4,200`$`, before debugging and hyperparameter tunning. This is far beyond our budget, which is already overspent, and in our opinion a far less valuable experimentation set.
>
> **Closing Remarks**
>
> As in [1] and [4], our experiments are comparative, and designed to ground our theoretical contributions rather than achieve SOTA performance. The reduced vanilla performance aligns with [4], and our results remain consistent with [1] and [2], as [1] did not test on ImageNet, and [2] did not compare with VIB or untargeted attacks on ImageNet. By adding modalities and attack methods, our work enhances and builds upon prior research while remaining fully consistent with it.
>
> Thank you again for your detailed review, responsiveness, and for the deep and enriching discussion.
>
> **References**
>
> [7] Training ImageNet on p3 instances https://aws.amazon.com/blogs/machine-learning/new-speed-record-set-for-training-deep-learning-models-on-aws/
>
> [8] Training BERT on p3 instances https://aws.amazon.com/blogs/machine-learning/amazon-web-services-achieves-fastest-training-times-for-bert-and-mask-r-cnn/

---

### Official Review · Reviewer_Hx2V · 2024-11-01

**Soundness:** 1
**Presentation:** 1
**Contribution:** 1
**Rating:** 3
**Confidence:** 5

**Summary:**

This paper revisits the variational information bottleneck and extends it to a supervised variational information bottleneck.  The experiment on ImageNet and text classification shows that SIB achieves better results than VIB.

**Strengths:**

SIB performs better than VIB regarding classification accuracy and adversarial robustness.

**Weaknesses:**

Overall, the novelty is insufficient, and the motivation is unclear. Some related works are missing.

### Novelty:
1. Using variational lower or upper bonds to approximate mutual information is not novel.
2. SIB's application focuses on traditional image classification and adversarial attacks. It doesn't include other applications like time series or more challenging scenarios like out-of-distribution or few-shot learning.
3. Compared to VIB, SIB adds $H( \hat{Y} \mid Z)$, involves new hyperparameters and new terms to approximate, and cannot guarantee to reach an accurate value for $H( \hat{Y} \mid Z)$.

### Motivation:
1. The title of the paper uses "revisiting," which refers to why the VIB is revisited and what the problem of VIB is.  These two parts are not clear in the paper. Figure 1 cannot demonstrate well since an overfitted decoder can also exist in SIB if the $\lambda$ is not set appropriately.

### Missing Reference:

[1] Kolchinsky, Artemy, Brendan D. Tracey, and David H. Wolpert. "Nonlinear information bottleneck." Entropy 21.12 (2019): 1181.

[2] A. Zaidi and I. E. Aguerri, “Distributed deep variational information bottleneck,” in Proc. IEEE 21st Int. Workshop Signal Process. Adv. Wirel. Commun., 2020, pp. 1–5

[3] S. Sinha, H. Bharadhwaj, A. Goyal, H. Larochelle, A. Garg, and F. Shkurti, “DIBS: Diversity inducing information bottleneck in model ensembles,” in Proc. AAAI Conf. Artif. Intell., 2021, pp. 9666–9674

[4]  S. Mai, Y. Zeng, and H. Hu, “Multimodal information bottleneck: Learning minimal sufficient uni modal and multimodal representations,” IEEETrans. Multimedia, vol. 25, pp. 4121–4134, 2022

[5] K. W. Ma, J. P. Lewis, and W. B. Kleijn, “The HSIC bottleneck: Deep learning without back-propagation,” in Proc. AAAI Conf. Artif. Intell., 2020, pp. 5085–5092.

The paper should compare the above methods as well and also put them into a related work section.

**Questions:**

1. Could you please provide more experiments on PGD and AutoAttack?
2. Could you please visualize the latent representations of SIB and compare them with VIB?

---

> ### Author Response · Authors · 2024-11-26
> **Rebuttal response**
>
> We thank the reviewer for their feedback. We're pleased the reviewer recognizes that SVIB outperforms VIB in terms of accuracy and robustness.
>
> We note that we've uploaded a revised version of our submission, the details of which are specified in the general comment.
>
> **The reviewer notes that the motivation of the paper is unclear**
>
> VIB [1] applies IB [2] in supervised settings, where the downstream RV $Y$ is unknown. This is not the case in the original IB, which performs unsupervised clustering [3]. In the original VIB derivation $Y$ is assumed to be known and hence ignored, when in practice it is optimized over. To resolve this duality, we propose a new adaptation of IB to supervised tasks by admitting a new RV $\tilde{Y}$, and a new bottleneck between $Z$ and $\tilde{Y}$. Apart from resolving the duality, our proposed adaptation extends the theoretical framework proposed in [4], by defining and mitigating overfitting in decoders.
>
> This motivation is brought forth throughout the paper, for example in lines 17-22, 54-62, 478-484, in Section 3.4 Motivation, and in the discussion. Following the reviewer's comment we've adjusted the abstract, and replaced the first paragraph of the intro to better and earlier convey the motivation of our work.
>
> **The reviewer notes that Figure 1 does not hold for all $\lambda$ values**
>
> The purpose of Figure 1 is to illustrate Eq 17. We didn't claim that it hold for all SVIB models, and we note that any value of $\lambda\ge0$ will increase decoder entropy. We ask the reviewer to clarify this remark if possible.
>
> **The reviewer notes that the paper lacks in novelty**
>
> We believe our paper introduces a new and better adaptation of IB to supervised DNNs, an extension [1] and [4], and a new information theoretical model for overfitting. Our method is presented bottom up, making the paper accessible to researchers outside the field, and laying down many possible avenues for future research.
>
> While our work does not include time series, out-of-distribution or few-shot learning, we feel that the provided experiments are more than sufficient to ground our theoretical approach. Most VIB papers deal with a single modality, and in many cases solve low dimensional data. Our work tackles high dimensional data of different modalities over 6 different tasks, with 5 experiments per run to attain statistical significance.
>
> We note that the goal of this paper is not to benchmark adversarial robustness or classification. We aim at a better adaptation of IB to classifier DNNs, and the theoretical insights that follow from this adaptation in the information theoretic models of DNNs.
>
> **The reviewer notes that SVIB introduces a new hyperparameter**
>
> SVIB shows consistent behavior over different $\lambda$ values, as shown in Appendix C.
>
> **The reviewer notes that $H(\tilde{Y}|Z)$ is not tight**
>
> We don't claim any tightness on $H(\tilde{Y}|Z)$, nor is it required for our purpose. Furthermore, such bounds are common practice in VIB literature. We ask the reviewer to clarify this remark if possible.
>
> **The reviewer notes missing references**
>
> We give a brief overview:
>
> - Nonlinear information bottleneck (Entropy 2019): Suggests using a non parametric MI estimator for I(X;Z).
>
> - Distributed deep variational information bottleneck (IEEE 2020) and DIBS (AAAI 2021): Both suggest an ensemble architecture for VIB.
>
> - Multimodal information bottleneck (IEEETrans 2022): Applies VIB to multimodal joint embedding problems.
>
> - HSIC bottleneck (AAAI 2020): Suggest an IB inspired statistical method to replace backpropagation.
>
> These papers are out of scope for our research, are irrelevant to the problem we are tackling in the work, are mostly inapplicable to our experimental setting, in 2 cases require special datasets or optimization method, and citing them will not contribute to the work, and will confuse readers.
>
> The Nonlinear information bottleneck is the only mentioned work that can realisticly be added to our experiments, yet it tackles the problem of mutual information estimation for the rate term, which is not relevant to our research, and can easily go hand in hand with SVIB, such as done in [5].
>
> **The reviewer asks for PGD and AutoAttack experiments**
>
> As we are limited in time and resources we cannot address all attacks. Our experiments already feature 5 different attacks on 2 modalities and 4 different model types. Moreover, PGD is akin to an iterative form of FGS which we perform, and scores on FGS are a good proxy for PGD.
>
> **The reviewer asks for latent plots of VIB and SVIB**
>
> We've added t-SNE plots for all 4 model types in Appendix C, as the reviewer suggested.
>
> **References**
>
> [1] Deep Variational Information 260 Bottleneck. ICLR 2017
>
> [2] The Information Bottleneck Method. Allerton 1999
>
> [3] The information bottleneck: Theory and applications. Hebrew University, 2002
>
> [4] Fixing a Broken ELBO. PMLR 2018
>
> [5] CLUB: A Contrastive Log-ratio Upper Bound of Mutual Information. ICML 2020

---

> ### Comment · Reviewer_Hx2V · 2024-11-27
>
> Thank you for your response. However, I still find the motivation and novelty of the proposed work unconvincing.
>
> ### Motivation
>
> **1.VIB applies IB in supervised setting not the original IB for the unsupervised clustering.**
>
> The proposed SVIB still focus on supervised tasks, which does not appear to address this limitation.
>
> **2. Preventing the classifier from overfitting to the representation by involving maximize  H(\hat{Y}|Z)**
>
> Why not directly maximize H(\hat{Y}|Z) within the standard VIB framework, rather than introducing an additional information bottleneck?
>
> ### Novelty
>
> **1. Most VIB papers deal with a single modality, and in many cases solve low dimensional data.**
>
> For multi-modal scenarios, [1] ("Multimodal Information Bottleneck," IEEE Trans, 2022) already addresses this challenge. I believe VIB itself is capable of handling multi-modality. Additionally, I am skeptical about the effectiveness of SVIB in high-dimensional settings since its Mutual Information (MI) estimation approach is the same as VIB's.
>
> **2. This work is an extension of [2] ("Fixing a Broken ELBO," PMLR 2018).**
>
> The contribution appears incremental.
>
> ### Missing References and Limited Scope.
>
> From my understanding, the primary contribution is the constrained maximization H(\hat{Y}|Z). However, there are other IB-based methods with different MI estimation perspectives. Why not evaluate the effectiveness of maximizing H(\hat{Y}|Z) in those frameworks to validate its broader applicability? If SVIB only modifies VIB and is constrained to variational lower-bound methods, the scope of the paper appears too limited.
>
> Based on the reasons outlined above, I will maintain my current score.

---

> ### Author Response · Authors · 2024-12-04
>
> We thank the reviewer for responding.
>
> **The reviewer claims that since SVIB is applied in supervised settings, it does not adapt IB to supervised settings**
>
> This claim is logically flawed, in order to adapt IB to supervised tasks, one must apply it in these settings.
>
> We recap the relaxation performed in VIB, and how SVIB amends it by adapting IB to supervised settings:
>
> - As stated above, and in detail in the paper, the IB [3] learns an unsupervised representation $Z$ under the assumption that the joint distribution $p(x,y)$ is known, the Markov chain Z-X-Y, and $p(z|x)$ as the only optimized parameter.
>
> - VIB [1] first adapted IB to supervised tasks by introducing the variational $c(y) = \int \int c(y|z) \, p(z|x) \, p(x) \, \mathrm{d}x \, \mathrm{d}z$ instead of the known $p(y)$.
>
> - VIB derives a variational upper bound to the IB, but relaxes the derivation by considering the RV $Y$ at times as the optimized variational $c(y)$, and at times as the empirically observed $\hat{Y}$.
>
> - [6] first observed that VIB adheres to the Markov chain $\tilde{Y}-Z-X-Y$, where $\tilde{Y}$ is the variational distribution introduced in VIB.
>
> - We gave a formal definition of $\tilde{Y}$, and reformulated the IB problem definition to adhere to the new Markov chain. We then derive variational bounds on our new formulation without relaxing the problem, distinguishing between the observed $Y$ and the optimized variational $\tilde{Y}$ in our problem definition and derivation.
>
> **The reviewer asks why not to maximize conditional entropy without the IB derivation**
>
> The motivation of the work is to advance the information theoretical study of DNNs using the IB framework.
>
> The conditional entropy regularization term emerges from our derivation as an upper bound to the second bottleneck. While previous work proposed the usage of similar terms [7,8], they did not provide the theoretical justification and corollaries.
>
> **The reviewer notes that our extension of [2] is incremental**
>
> One of the main claims of [2] is that the ELBO loss function is prone to overfitting representations. Since VIB is equivalent to the ELBO, this applies to VIB as well.
>
> When considering the Markov chain $\tilde{Y}-Z-X-Y$ proposed in [6], it is evident that minimizing $I(X;Z)$ while maximizing $I(Z;\tilde{Y})$ when only $X$ is constant, can easily reach the trivial solution of $I(X;Z)=0$, which is not the case in the unsupervised IB setting, where $p(x,y)$ is known and given, and only $p(z|x)$ is optimized over.
>
> Better applying IB on this problem results in the admission of a second bottleneck, variationally bounded by $H(\tilde{Y}|Z)$, preventing the collapse to the trivial solution.
>
> Our work represents a significant advancement over [2], offering a theoretical resolution to one of its central claim. We ask the reviewer to consider for what reasons did he find our extension incremental, and what would he consider a non incremental improvement.
>
> **Mulitmoality**
>
> We agree that VIB can successfully deal with multimodal problems, as shown in [9]. But we don't understand the relevance of this claim to our study.
>
> **Applying different MI estimation methods**
>
> Indeed there are many MI estimation methods apart from the ones applied in [1]. Yet, our study deals with the reformulation of the IB problem to supervised tasks, and does not study the precision of mutual information estimation.
>
> Thank you again for your time and effort.
>
> **References**
>
> [6] Bernhard Geiger and Ian Fischer. A comparison of variational bounds for the information bottleneck functional.
>
> [7] Rethinking the inception architecture for computer vision. IEEE 2016.
>
> [8] Regularizing neural networks by penalizing confident output distributions. ICLR 2017
>
> [9] Multimodal information bottleneck. IEEETrans 2022

---

### Official Review · Reviewer_LjHJ · 2024-11-04

**Soundness:** 1
**Presentation:** 1
**Contribution:** 2
**Rating:** 3
**Confidence:** 3

**Summary:**

The paper claims to provide the "theoretical optimal approach to data modeling", and to extend the framework, by deriving the a variational bound to resolve some problems of the previous framework.

My understanding is that eq. 16 (derived by eq.3)  is the contribution of this work. The paper provides derivations to justify eq. 16.

In the experimental session, the authors evaluate the performance in image and text classification of the new loss and the robustness to adversarial attack.

The authors claim that the new loss outperforms the previous loss.

**Strengths:**

If the paper were clearer, the paper presents many contributions and analyses of the proposed terms.

Two experiments (image and text classification) compare the "vanilla" and VIB with the new loss.

Historical overview of IB.

**Weaknesses:**

The paper is largely unclear.

It starts with the history of the IB, more than presenting the contribution of the work.

The presentation is not clear on the steps of the new loss.

In the new loss, there seems to be a new contribution on the predictor, but the predictor (or classifier) is already included in the loss in the VIB.

The impression is that the new loss introduces a new regularization term, but its justification is not clear.

The abstract is unclear, what are the two points of the "dual role"? What is the "theoretically optimal approach to data modeling"?

**Questions:**

Would be nice to understand the difference of eq. 16 and the standard VIB.

**Details Of Ethics Concerns:**

No ethics is foreseen in this work.

---

> ### Author Response · Authors · 2024-11-26
> **Rebuttal response**
>
> We thank the reviewer for their feedback, it's encouraging to hear that the reviewer finds many contributions in our work.
>
> We note that we've uploaded a revised version of our submission, the details of which are specified in the general comment.
>
> **Clarity of presentation and contribution**
>
> The current study aims at better adapting IB [1] and VIB [2] to supervised tasks: VIB [1] applies IB [2] in supervised settings, where the downstream RV $Y$ is unknown. This is not the case in the original IB, which performs unsupervised clustering [3]. In the original VIB derivation $Y$ is assumed to be known and hence ignored, when in practice it is optimized over. To resolve this duality, we propose a new adaptation of IB to supervised tasks by admitting a new RV $\tilde{Y}$, and a new bottleneck between $Z$ and $\tilde{Y}$. Apart from resolving the duality, our proposed adaptation extends the theoretical framework proposed in [4], by defining and mitigating overfitting in decoders.
>
> This motivation is brought forth throughout the paper, for example in lines 17-22, 54-62, 478-484, in Section 3.4 Motivation, and in the discussion. Following the reviewer's comment we've adjusted the abstract, and replaced the first paragraph of the intro to better and earlier convey the motivation of our work.
>
> **IB Optimality**
>
> Representations learned with the IB method are largely considered optimal given the implicit assumptions [3]:
>
> - Optimizing a precision-complexity trade-off will yield a model that is closer in nature to the real underlying process
>
> - Mutual information is a sufficient metric for this purpose.
>
> Following the reviewer's remark, we've added these caveats as a footnote in the introduction.
>
> We note that we do not consider our variational approach to IB as optimal, nor that we claim this in the paper.
>
> **The reviewer notes that the derivation steps are unclear**
>
> We made our best efforts to make the derivation as clear as possible, by being verbose about each step and giving clear definitions, both in the paper body, and in the preliminaries in Appendix A. We find that there is no 'one size fits all', and that some readers find verbose writing unclear, while others find minimalist writing unclear. We ask the reviewer to clarify which part of the derivation did he found unclear.
>
> **The reviewer notes that the predictor is already derived in VIB**
>
> Indeed, VIB [2] takes into account the RV $Y$ in the course of derivation, but performs a relaxation by treating $Y$ both as the variational discriminator, and the as the presumably known RV. Consider the VIB loss as derived in Eq. (15) at [2]:
>
> $$\mathcal{L}_{VIB}\equiv\int\int\int p(x)p(y|x)p(z|x)\log{q(y|z)})\mathrm{d}x\mathrm{d}y\mathrm{d}z-\beta \int\int\int p(x)p(z|x)\log {\left(\frac{p(z|x)}{r(z)}\right)}\mathrm{d}x\mathrm{d}z$$
>
> We have that $Y$ is represented both as the variational $q(y|z)$, and as the presumably known, and later empirically sampled, $p(y|x)$. This duality does not exist in the original IB paper, which performs unsupervised clustering [3]. Our work aims at better adapting IB to supervised tasks by admitting the variational $\tilde{Y}$ to the original problem formulation, together with an additional bottleneck, as stated in Eq (2). As is common in the field, we bound our new bottleneck and derive a tractable variational approximation in Equations (9-13), resulting in a new scaled conditional entropy term, which we append to the original VIB objective in Eq (15):
>
> $$\lambda H(\tilde{Y}|Z) = \lambda \int\int\int p(x)e(z|x)c(y|z)log\left(c(y|z)\right)\mathrm{d}x\mathrm{d}y\mathrm{d}z$$
>
> Interestingly, we find that our method provide a possible remedy to the discrepancies in the VIB and ELBO loss discussed in [4], a well received and influential paper, by providing decoder regularization.
>
> Following previous notes by the reviewer, we've amended the introduction and abstract to convey this notion earlier and in a clearer manner in the paper.
>
> **The reviewer asks for clarification regarding the difference between Eq (16) and VIB**
>
> Empirically, Eq (16) is equivalent to VIB together with the aforementioned scaled conditional entropy term. The theoretical motivation was elaborated above.
>
> **References**
>
> [1] The Information Bottleneck Method. The 37th annual Allerton Conference 1999
>
> [2] Deep Variational Information 260 Bottleneck. ICLR 2017
>
> [3] The information bottleneck: Theory and applications. Hebrew University, 2002
>
> [4] Fixing a Broken ELBO. PMLR 2018

---

### Author Response · Authors · 2024-11-26
**Details of the new revision**

Based on the given feedback, we've submitted a revised version with the following changes:

- Added extensive experiments for CEB [1], elaborated on [1] in the related work section, and added our CEB implementation to the attached code.

- Doubled the amount of VIB [2] experiments, which now cover the entire range of useful $\beta$ values.

- Replaced Tables 1, 2 with plots. Please note that all plots and tables now show 'higher is better $\uparrow$' scores, for better readability.

- Edited the Venn diagram in Figure 1 (Now Figure 2) to adhere to the Markov chain $Y-X-Z-\tilde{Y}$.

- Removed $p(x,y)$ as arguments in the problem definition (Equation 2).

- Revised the abstract and first paragraph of the introduction to better, and earlier, convey our main contribution and motivation.

- Added a footnote in the introduction qualifying the assumptions under which the IB method [4] is considered optimal.

- Removed Eq (7) by directly stepping from Eq (6) to Eq (8) by citing the Barber-Agakov inequality [3], to produce a cleaner derivation.

- Added t-SNE visualization of the latent space for all 4 models in Appendix C, together with a short interpretation of our findings.

We feel that the revised paper conveys a much clearer message, and provides stronger and more accessible empirical grounding for our claims. We thank the reviewers for their valuable and detailed feedback, which made this possible.

We hope that our discussion in the replies, and the submitted changes, have addressed all concerns. We look forward to answer any additional remarks, and to continue the discussion.

**References**

[1] CEB Improves Model Robustness, Entropy 2020.

[2] Deep Variational Information 260 Bottleneck. ICLR 2017

[3] Barber, D.; and Agakov, F. V. 2003. The IM algorithm: a variational approach to Information Maximization. In Neural Information Processing Systems.

[4] The Information Bottleneck Method. The 37th annual Allerton Conference 1999

---

### Meta-Review · Area_Chair_5Efo · 2024-12-21

**Metareview:**

The paper proposes a modification of the Information Bottleneck framework intended to address an overfitting issue identified in prior work. They use the framework to derive a modified supervised training objective, and include experiments showing improvements over VIB in certain settings.

Reviewers appreciated the motivation to revisit tractable IB objectives. However, reviewers had significant concerns about the clarity of presentation, and about the strength of both the theoretical and experimental components.
After reading the paper, I agree with this assessment, and thus I must recommend rejection at this time.

**Additional Comments On Reviewer Discussion:**

See above

---

### Decision · Program_Chairs · 2025-01-22

Reject